# Unifying the Perspectives of NLP and Software Engineering: A Survey on Language Models for Code

**Ziyin Zhang**
*Shanghai Jiao Tong University*                                                  *daenerystargaryen@sjtu.edu.cn*
*Ant Group*

**Chaoyu Chen**
*Ant Group*

**Bingchang Liu**
*Ant Group*

**Cong Liao**
*Ant Group*

**Zi Gong**
*Ant Group*

**Hang Yu**[*]                                                                *hyu.hugo@antgroup.com*
*Ant Group*

**Jianguo Li**[*]                                                               *lijg.zero@antgroup.com*
*Ant Group*

**Rui Wang**[*]                                                               *wangrui12@sjtu.edu.cn*
*Shanghai Jiao Tong University*

**Reviewed on OpenReview:** *https://openreview.net/forum?id=hkNnGqZnpa*

## Abstract

In this work we systematically review the recent advancements in software engineering with language models, covering 70+ models, 40+ evaluation tasks, 180+ datasets, and 900 related works. Unlike previous works, we integrate software engineering (SE) with natural language processing (NLP) by discussing the perspectives of both sides: SE applies language models for development automation, while NLP adopts SE tasks for language model evaluation. We break down code processing models into general language models represented by the GPT family and specialized models that are specifically pretrained on code, often with tailored objectives. We discuss the relations and differences between these models, and highlight the historical transition of code modeling from statistical models and RNNs to pretrained Transformers and LLMs, which is exactly the same course that had been taken by NLP. We also go beyond programming and review LLMs' application in other software engineering activities including requirement engineering, testing, deployment, and operations in an endeavor to provide a global view of NLP in SE, and identify key challenges and potential future directions in this domain. We keep the survey open and updated on GitHub at https://github.com/codefuse-ai/Awesome-Code-LLM.

---

[*]Corresponding authors.

# 1 Introduction

Language modeling has advanced remarkably in recent years with the advent of pretrained Transformers (Vaswani et al., 2017) such as BERT (Devlin et al., 2019) and GPT (Radford et al., 2018). As large language models (LLMs) scaled to hundreds of billions of parameters and started to display early signs of artificial general intelligence (Brown et al., 2020; Chowdhery et al., 2023; OpenAI, 2023), their applications have also transcended text processing. Pioneered by Codex (Chen et al., 2021b), LLMs have achieved impressive results in code processing, giving rise to commercial products such as GitHub Copilot[1] and open-source multi-billion code models such as StarCoder (Li et al., 2023i) and Code LLaMA (Rozière et al., 2023).

The application of pretrained Transformers in software engineering, however, can be traced back to dates before decoder-only autoregressive models became dominant (Feng et al., 2020; Liu et al., 2020), and also goes beyond code processing (Arora et al., 2023; Feng et al., 2024; Ma et al., 2024). However, this domain is yet to witness a comprehensive review. In an attempt to bridge the gap between natural language processing (NLP) community and software engineering (SE) community on the topic of language model applications, we undertake a panoramic survey of language models for SE in this work, covering 70+ models, 40+ downstream tasks, 180+ datasets, and 900 related works. Putting the emphasis on code language models, we analyze their development, discuss their differences from general language models, and highlight the integration of code-specific features such as abstract syntax trees or data flows, as well as the latest techniques adapted from NLP.

Related to our work, we are aware of several surveys on similar topics, with three works concurrent to us (Hou et al., 2023; Zheng et al., 2023b; She et al., 2023). These works, however, focus either on NLP side (Zan et al., 2023; Xu & Zhu, 2022) or SE side (Niu et al., 2023; Hou et al., 2023; Zheng et al., 2023b; She et al., 2023), and do not cover models, tasks, and challenges from the other side. For example, Zan et al. (2023) focus on LLMs for text-to-code generation, while giving little discussion of other applications in software engineering community. Hou et al. (2023) and She et al. (2023), in contrast, comprehensively review works from SE venues such as ASE and ICSE, but cite only a handful of works from deep learning and NLP venues such as ACL, EMNLP, NeurIPS, and ICLR.

Thus, building on these works, we endeavor to unite the perspectives from both communities, and highlight the relationship between NLP and SE: SE applies language models to various tasks for automation, while NLP adopts tasks from SE to evaluate language models. We observe that advanced topics from language modeling have been recently introduced into code processing, including instruction tuning (Honovich et al., 2023; Xu et al., 2024; Luo et al., 2023), infilling objectives (Tay et al., 2023b; Li et al., 2023i; Rozière et al., 2023), recontemplation of scaling laws (Hoffmann et al., 2022; Gunasekar et al., 2023; Li et al., 2023j), architectural improvements (Shazeer, 2019; Su et al., 2024; Dao et al., 2022), and autonomous agents (Qian et al., 2023; Hong et al., 2023), while in return SE requirements, represented by programming (Chen et al., 2021b), are providing real-world testbeds for these technologies and driving the development of LLMs forward into production and deployment. We believe a systematic review of these advancements would benefit both communities. Furthermore, unlike the existing reviews that focus on programming-related tasks, we are also the first to explicitly go beyond programming and provide a global view of LLM applications in the full life cycle of software development, covering distinct stages such as requirement engineering, deployment, and operations.

The rest of this work is organized following the taxonomy presented in Figure 1. In Section 2 we first contextualize the downstream tasks in software engineering, highlighting the focus on programming related tasks. Then, in Section 3 we provide the preliminaries of language modeling and Transformer models, and in Section 4 we discuss the plethora of LLMs that have demonstrated coding ability. In Section 5 we review the specialized and often smaller models by their architecture, with special attention on the recent application of infilling objectives, instruction tuning, reinforcement learning, and engineering improvements. Then, in Section 6, we discuss unique features of code that are not available to natural languages but have been utilized to aid code processing. In Section 7, we review the most recent integration between LLMs and software development, before finally concluding this work in Section 8 and highlighting the current challenges in code processing.

---

[1] https://github.com/features/copilot

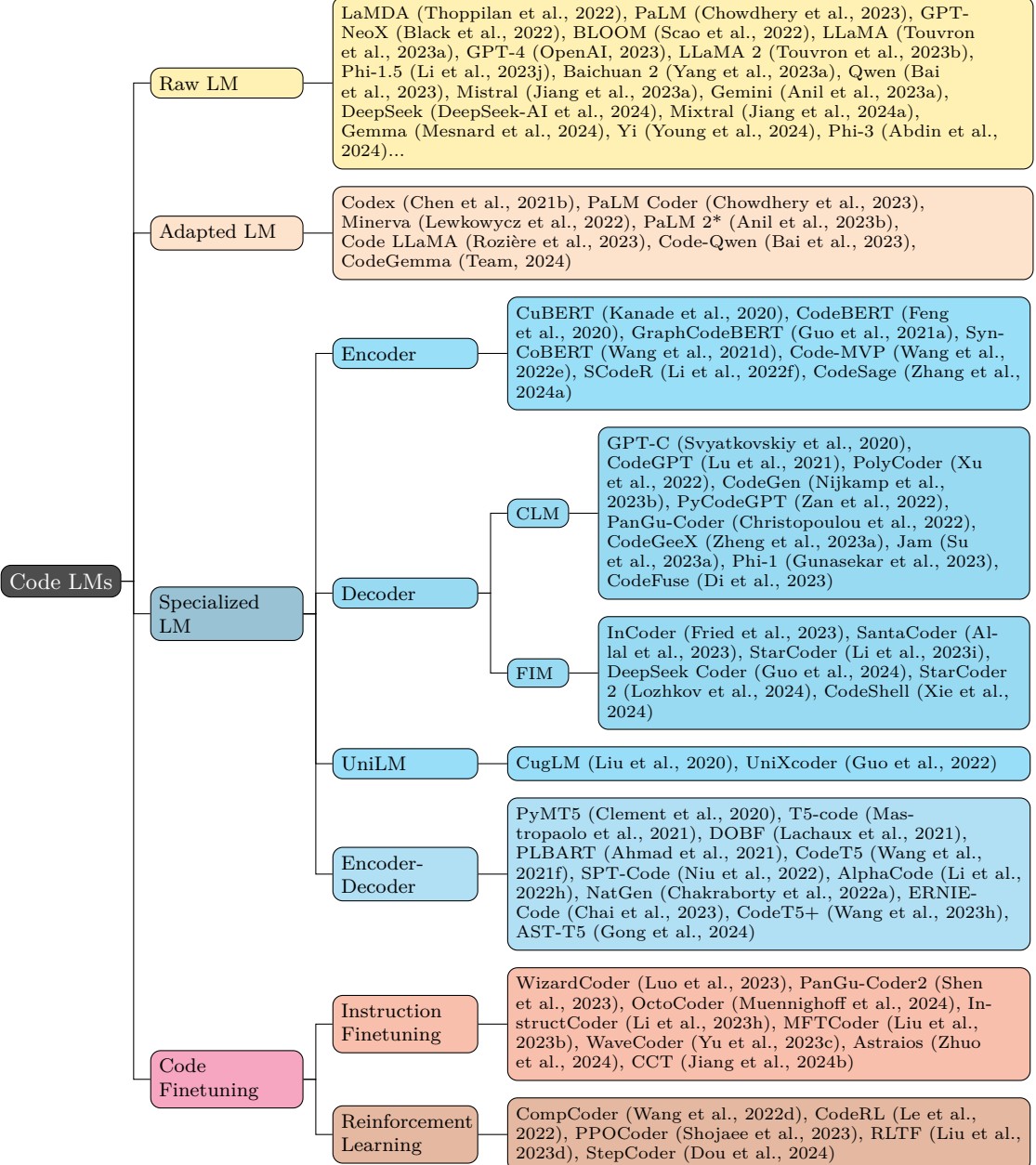

Figure 1: Our taxonomy of pretrained language models for code.

## 2 Downstream Tasks in Software Engineering

Over the past decade, the SE community has found applications of language models in various downstream tasks. CodeXGLUE (Lu et al., 2021) consolidates most of the code-related tasks into a single benchmark, while HumanEval Chen et al. (2021b) brought NL-to-code synthesis into the spotlight in the NLP community, which has since become a standard task for evaluating LLMs (Figure 2). However, other tasks, especially those not directly related to coding, have remained understudied.

In this section, we first briefly introduce each of the traditional SE tasks and the application of pretrained language models in them in 2.1, and provide a comprehensive list of related works for each task. Then, we review the evaluation metrics in 2.2 and investigate program synthesis in more detail in 2.3. Lastly, we

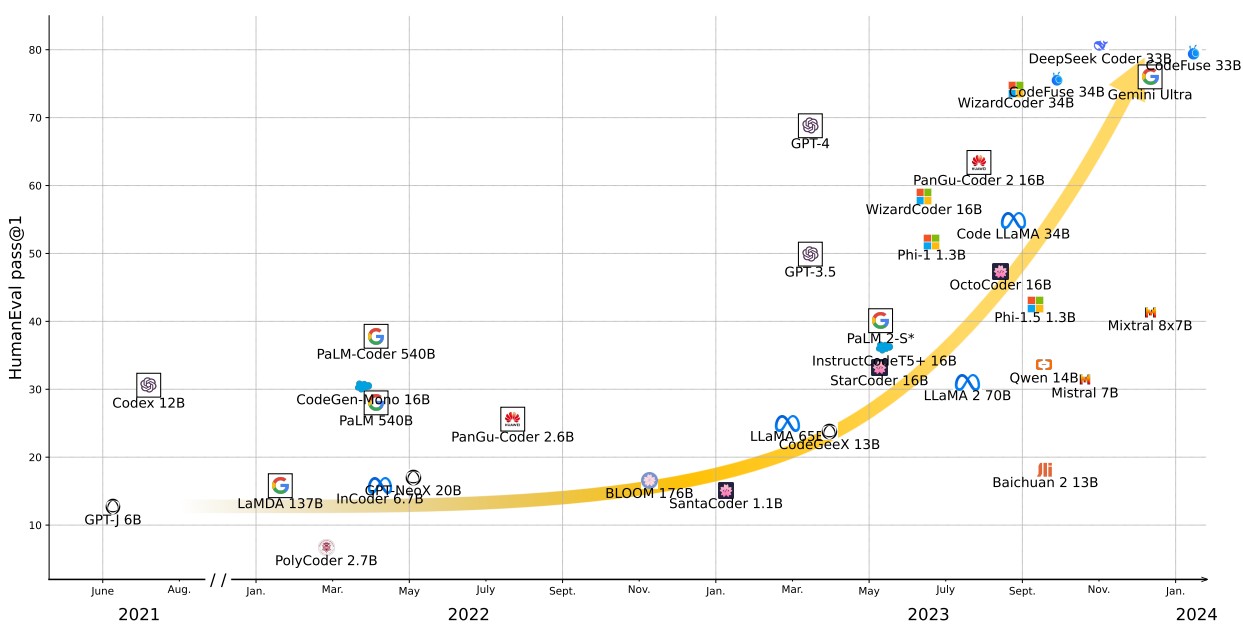

Figure 2: The timeline of code language models' progress on HumanEval.

also discuss the latest trend of repository-level coding in 2.4. In Appendix A and B respectively, we list benchmarks for each downstream task and language models' performance on them.

## 2.1 Downstream Tasks in SE

The custom in software engineering is to categorize downstream tasks according to their input/output modality, such as NL-to-PL (also referred to as text-to-code) tasks and PL-to-NL (i.e. code-to-text) tasks (Lu et al., 2021). However, as we are dedicated to the full life cycle of software development, we adopt a different taxonomy in this work, and classify downstream tasks according to the stages in software development: requirement engineering (2.1.1), development (2.1.2), testing and analysis (2.1.3), deployment and operations (2.1.4), and lastly several novel tasks recently proposed for evaluating LLMs (2.1.5). We note that this taxonomy is interleaved with the understanding-generation dichotomy in NLP, since each category may contain both understanding and generation tasks, as discussed in 2.1.6.

### 2.1.1 Requirement Engineering

Requirement engineering refers to the specification, analysis, and validation of software requirements, software modeling, and UI/UX design. Related works are listed in Figure 3.

- *Requirement analysis* refers to the process of extracting, summarizing, classifying, and validating software requirements. Most early works in this field rely on statistical NLP technologies such as POS (Part-of-Speech) tagging and dependency parsing to extract actions from requirements, while more recent works utilize LLMs to directly classify, summarize, or extract requirement clauses. Zhao et al. (2020) provide a fine-grained survey on requirement analysis.

- *UI/UX Design*, short for User Interface and User Experience design, is a fundamental step in software development. While many works on UI design utilize computer vision technologies for layout design (Cheng et al., 2023; Kulkarni et al., 2023; Feng et al., 2023b), we only focus on the intersection of this task and NLP technologies, such as using language models to generate markup languages.

- *Model generation* aims to generate software models in modeling languages such as UML (Unified Modeling Language). Abdelnabi et al. (2021a) and Ahmed et al. (2022) provide comprehensive reviews on this task before the rise of LLMs.

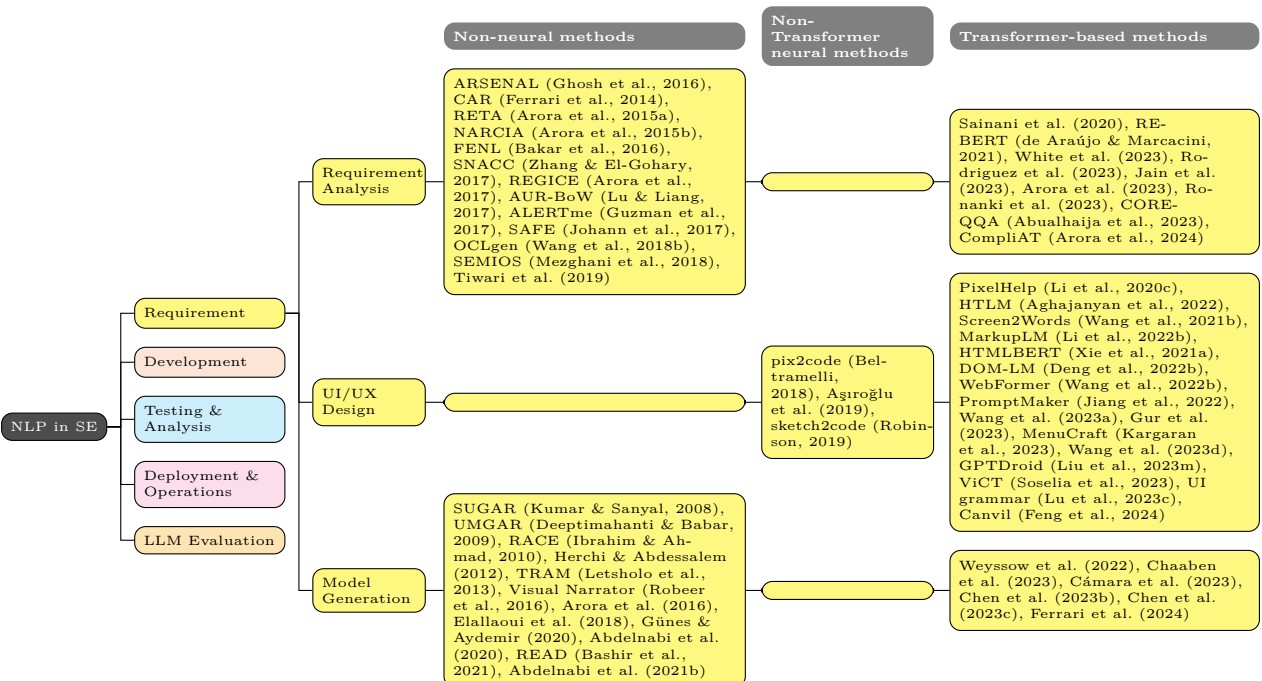

Figure 3: NLP applications in the *requirement engineering* stage of SE.

### 2.1.2 Development

The development phase is the stage where most programming activities take place, and also the stage where LLM applications are most abundant. We further categorize this stage into different activities such as code search, code generation, code editing, code suggestion, and code explanation. Related works are listed in Figure 4 and 5.

**Code Search**

- *NL-to-code search*, also referred to as text-to-code search, aims to retrieve relevant code given natural language queries, or to mine parallel text-code pairs from an unannotated corpus. This task is usually performed by computing a similarity metric between the embedding of query and candidate code, and the contextual embeddings produced by bidirectional language models - such as BERT - has proven to be extremely helpful. Grazia & Pradel (2023) and Xie et al. (2023a) provide comprehensive reviews on this topic.

- *Code-to-code search* is a similar task where the input is an existing code snippet, often in a different programming language from the target. Code-to-code search can be reformulated as finding clones of the query in the pool of targets, and is thus equivalent to clone detection to some extent.

- *API mining* refers to the process of finding similar APIs in different libraries, potentially in different programming languages. API mining is traditionally tackled by computing similarity metrics between source and target APIs using information retrieval models, but as generative models become ever more capable, it is also worth exploring to directly generate the target API as a sequence-to-sequence task. Another closely related task is *idiom mining* (Allamanis & Sutton, 2014), where the objective is to discover commonly used code patterns, which exposes the potential need for new APIs (Sivaraman et al., 2022).

**Code Generation**

- *Program synthesis* aims to generate code (usually a function or a method) given a natural language description. This task can be viewed as an updated version of NL-to-code retrieval using generative models instead

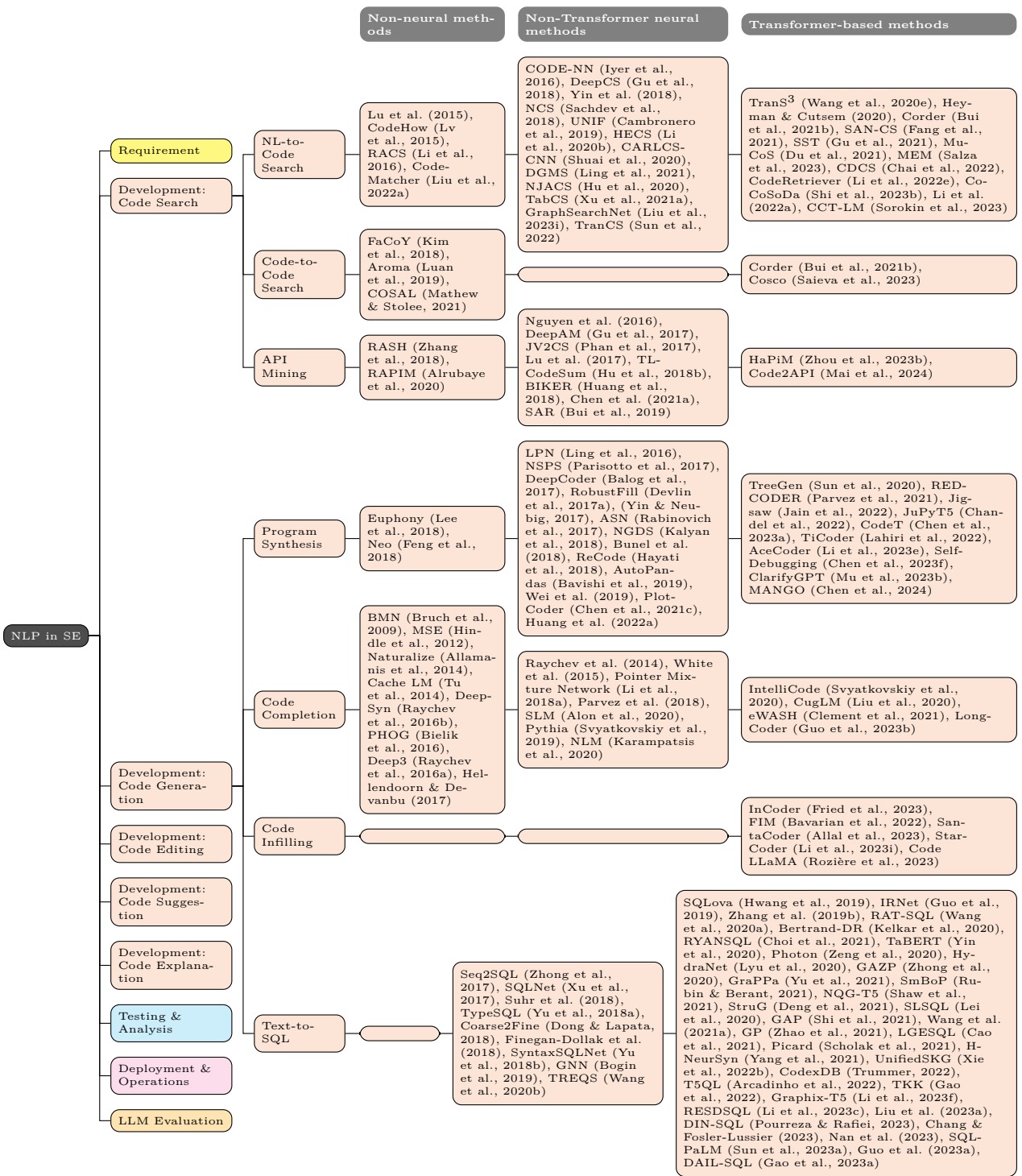

Figure 4: NLP applications in the *development* stage of SE.

of retrieval models. Statistical machine translation (SMT) and neural machine translation (NMT) models have been widely adopted for this task, often with enhanced decoders that leverage the unique grammatical rules of programming languages (Yin & Neubig, 2017; Rabinovich et al., 2017). Pretrained language models based on Transformer architecture, however, changed the game by directly generating the source code in

the autoregressive language modeling style, even without task-specific finetuning (Chen et al., 2021b). We discuss this task in more detail in Section 2.3.

- *Code completion* aims to complete a piece of code given its prefix, and remains to date one of the most popular applications of code language models in IDEs. This is essentially language modeling applied to code (which we dub "code modeling"), and related technologies have been progressively introduced: n-gram, RNN, and Transformer. However, due to the structured nature of programming languages, many early works found grammar-aided statistical models to perform better (Bielik et al., 2016; Hellendoorn & Devanbu, 2017), and neural models only became dominant after 2018.

- *Code infilling* is another recently proposed task, after fill-in-the-middle pretraining (Bavarian et al., 2022) became popular. It is a generalization of code completion, where not only the left context but also the right context is given.

- *Text-to-SQL* is a special case of program synthesis, where the model is tasked to generate SQL commands from natural language queries. It has been a topic of special interest in the NLP community (as can be seen from Figure 4), and one hypothesis for this is that SQL's declarative nature and restricted grammar make it easier to optimize compared with imperative programming languages such as C and Python (Marcus et al., 2019; 2020). We refer to Kumar et al. (2022); Deng et al. (2022a); Qin et al. (2022a); Katsogiannis-Meimarakis & Koutrika (2023) for surveys on this topic.

### Code Editing

- *Code translation* aims to translate a piece of code (usually a function or method) into another programming language. The relation between code translation and cross-lingual code search is similar to the one between program synthesis and text-to-code retrieval, and SMT/MNT models have also been widely applied to this task. One of the important applications of code translation is migrating old projects written in obsolete languages. However, we are yet to witness such applications at scale in the LLM era, as the context window of even the most powerful language models are quite limited in the face of such projects. Malyala et al. (2023) provide a short survey on this task from the SE perspective.

- *Program repair*, also known as bug fix, aims to fix a piece of buggy code. Like code translation, it is a traditional sequence-to-sequence generation task, and surveys are abundant on this topic (Gazzola et al., 2018; Monperrus, 2018; Zhong et al., 2022; Zhang et al., 2023d; Huang et al., 2023a).

### Code Suggestion

- *Type prediction* aims to predict the type of variables in dynamic programming languages such as Python and JavaScript. It has been used as a pretraining objective for code language models (Wang et al., 2022e), where it is often simplified as a binary tagging task to predict which tokens in the code are identifiers (Wang et al., 2021d;f).

- *Identifier prediction* is the task of predicting identifier names in the code. As these names are deemed to contain important semantic information, this task has been utilized for code summarization (Allamanis et al., 2016b), as well as pretraining code models (Wang et al., 2021f; Niu et al., 2022). A special case of identifier prediction is *method name prediction*.

- *Cloze test* is a recently proposed task for code understanding, after the rise of BERT-style pretraining. Due to the unique semantics of programming languages, several keywords are often selected for this test, such as `min` and `max` (Feng et al., 2020).

### Code Explanation

- *Code summarization*, *comment generation*, and *code documentation* all aim to generate explanations for code to facilitate understanding of the code, but with slightly different emphasis. Code summarization generates a natural language summary of the given code, while comment generation generates comments. These two terms are often used interchangeably in the literature, and their intended audience is usually developers. Code documentation, on the other hand, is more structured, includes more detailed information

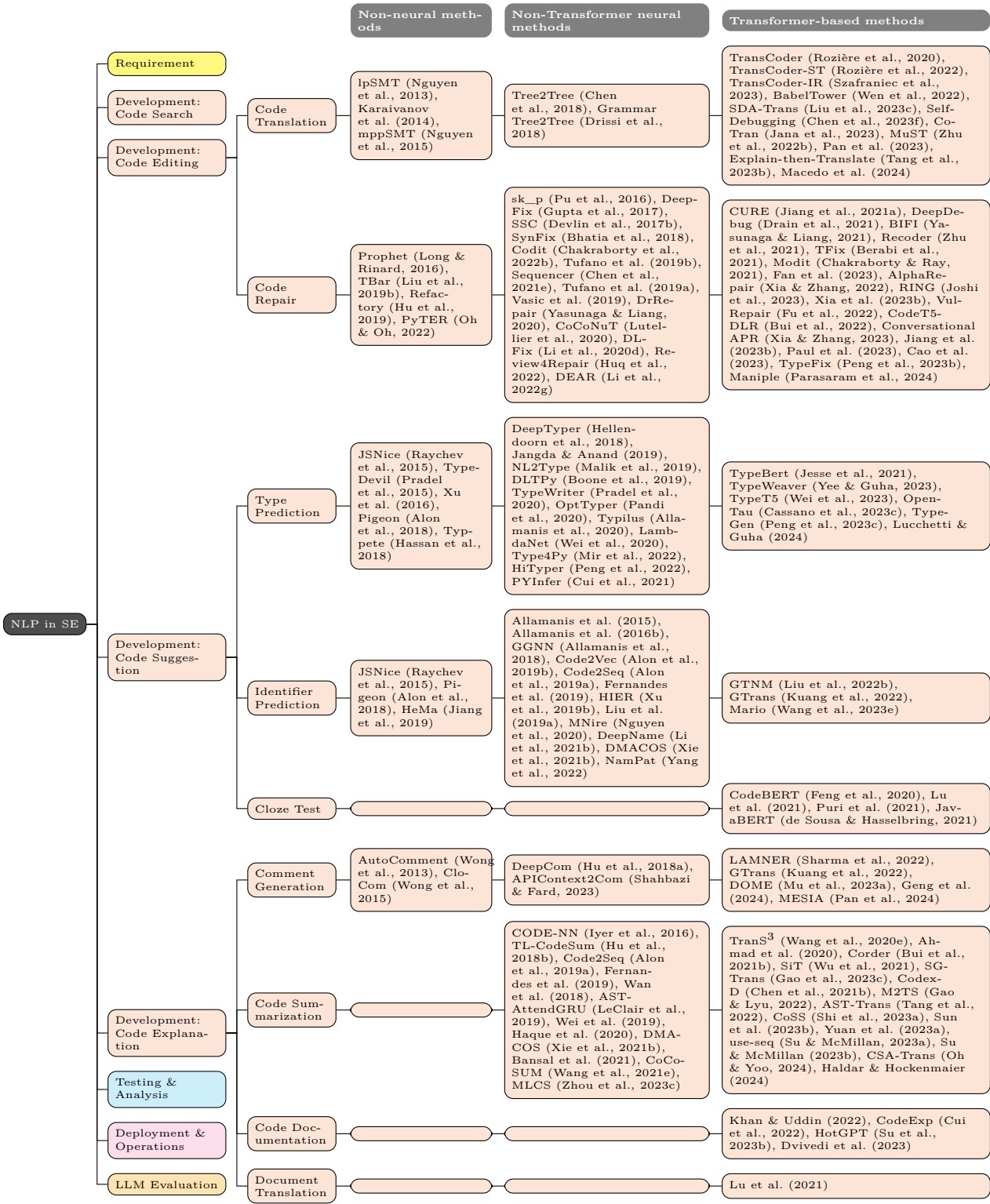

Figure 5: NLP applications in the *development* stage of SE.

about the code's usage and functions, and is usually intended for a broader audience. Zhang et al. (2022) provide a survey on code summarization.

- *Document translation* is the automatic translation of code-related documents. Since models, datasets, and prompting strategies for machine translation are abundant in NLP (Vaswani et al., 2017; Goyal et al., 2022; He et al., 2023b), we do not go into detail about this task.

### 2.1.3   Testing and Analysis

The third stage in software development is about testing and analyzing the correctness of programs (as well as other properties, such as security). Related works are listed in Figure 6.

**Testing**

- *Unit test generation* aims to generate unit tests for a given program. Prior to the rise of Codex and other code LLMs, almost all works in this area employed non-neural methods (see Figure 6). In the age of LLMs, however, this task is ever more important, as research has shown that the current unit tests for evaluating LLMs' program synthesis capability may be insufficient (Liu et al., 2023f). Wang et al. (2023b) provide a comprehensive survey on software testing with LLMs.

- *Assertion generation* is a subtask of unit testing. Given a program and a partial unit test, this task aims to generate assertions (also known as *oracles* in software engineering) within the unit test. This task has generally went unnoticed by the NLP community, as the program synthesis task used for evaluating LLMs often concern standalone, competition-style methods, for which the simple assertion of the equality between program output and expected answer suffices.

- *Mutant generation* aims to generate mutants of a given program for the purpose of mutation testing, and relates closely to unit test generation and assertion generation. A mutant that is not detected by a given set of unit tests and assertions indicates that either additional test cases or better assertions are required (Fraser & Arcuri, 2011). Recently, masking out tokens in the source code and sampling them from the output of a masked language model has become a common method for this task. Ojdanic et al. (2021; 2023) give empirical comparisons between different mutation methods.

- *Fuzzing* is another software testing task, where the objective is to generate a large set of inputs covering as many corner cases as possible. While many recent works on fuzzing target deep learning libraries, few have utilized language models to conduct this process (see Figure 6).

**Analysis**

- *Defect detection*, or *vulnerability detection*, predicts whether the input code is buggy or not, and is a standard single-sentence classification task. Nong et al. (2023); Steenhoek et al. (2023a); Bi et al. (2023); Harzevili et al. (2023); Zhou et al. (2024) provide surveys on this task.

- *Malware detection* is a similar task to defect detection. However, malware differs from other types of vulnerable code in that the bugs therein are malicious - i.e. they are intentionally injected. We refer to Sahin & Bahtiyar (2020) and Gopinath & Sethuraman (2023) for reviews on non-Transformer based methods for this task.

- *Clone detection* predicts whether or not two pieces of code are clones of each other. In software engineering there exist four types of code clones, and the most challenging type to identify is semantic clones, i.e. syntactically dissimilar code that have the same functionality. As this task can be viewed as a two-sentence classification task, BERT-style language models have been widely applied to it. Svajlenko & Roy (2020) and Zhang & Sakurai (2021) provide comprehensive reviews on non-deep-learning based methods for this task.

- *Code classification* aims to predict the functionality of a piece of code within a predefined set of labels. A very similar task is *author identification*, which predicts the author of the input code. Both tasks are standard single-sentence classification tasks, and traditional machine learning methods have been widely adopted in them (Kalgutkar et al., 2019), while pretrained language models have seen almost no application.

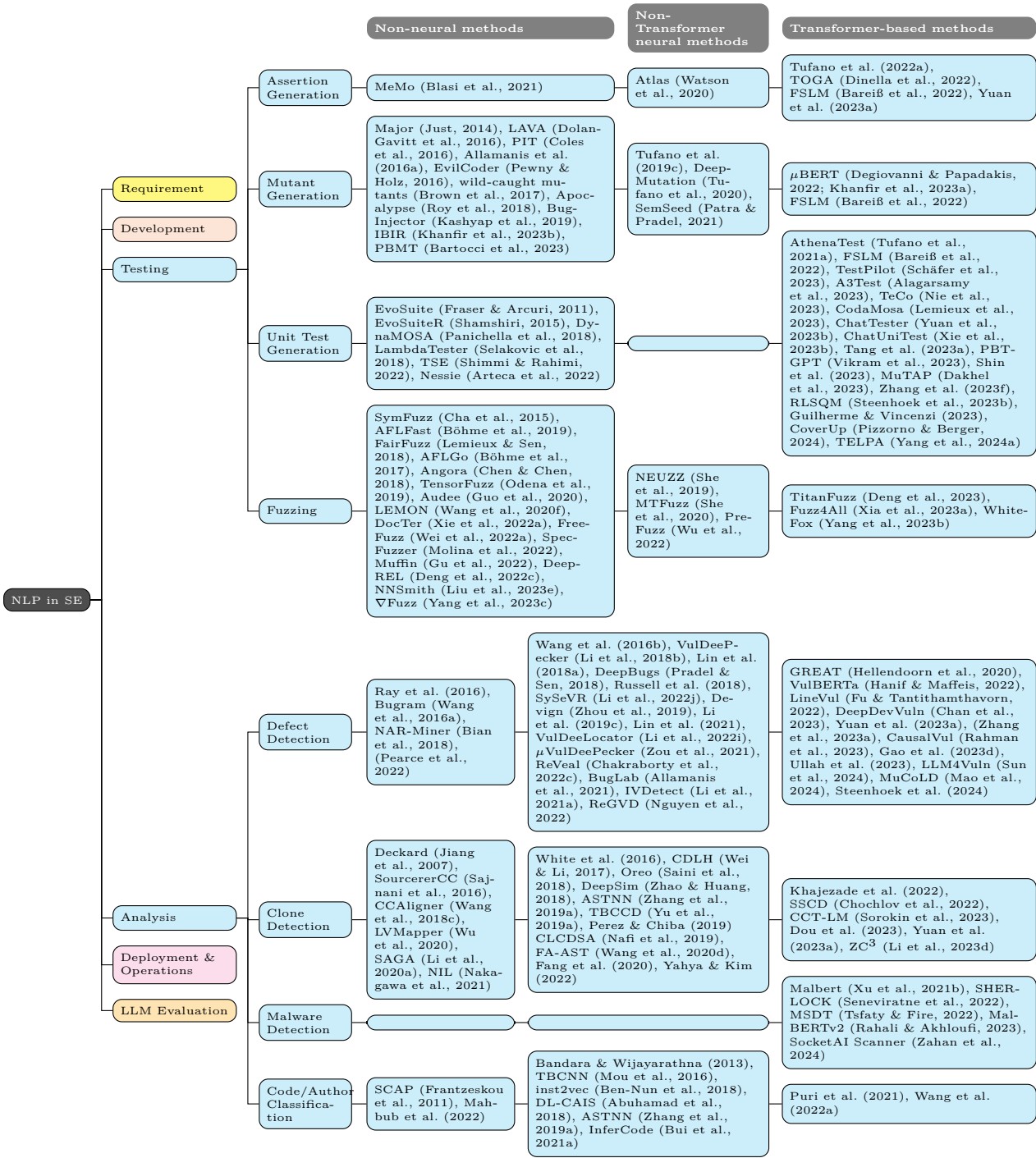

Figure 6: NLP applications in the *testing & analysis* stage of SE.

### 2.1.4 Deployment and Operations

After software passes the testing stage, it is deployed into production, and often continuously maintained or updated. For this stage, we summarize the specific tasks and their related works in Figure 7.

**Deployment**

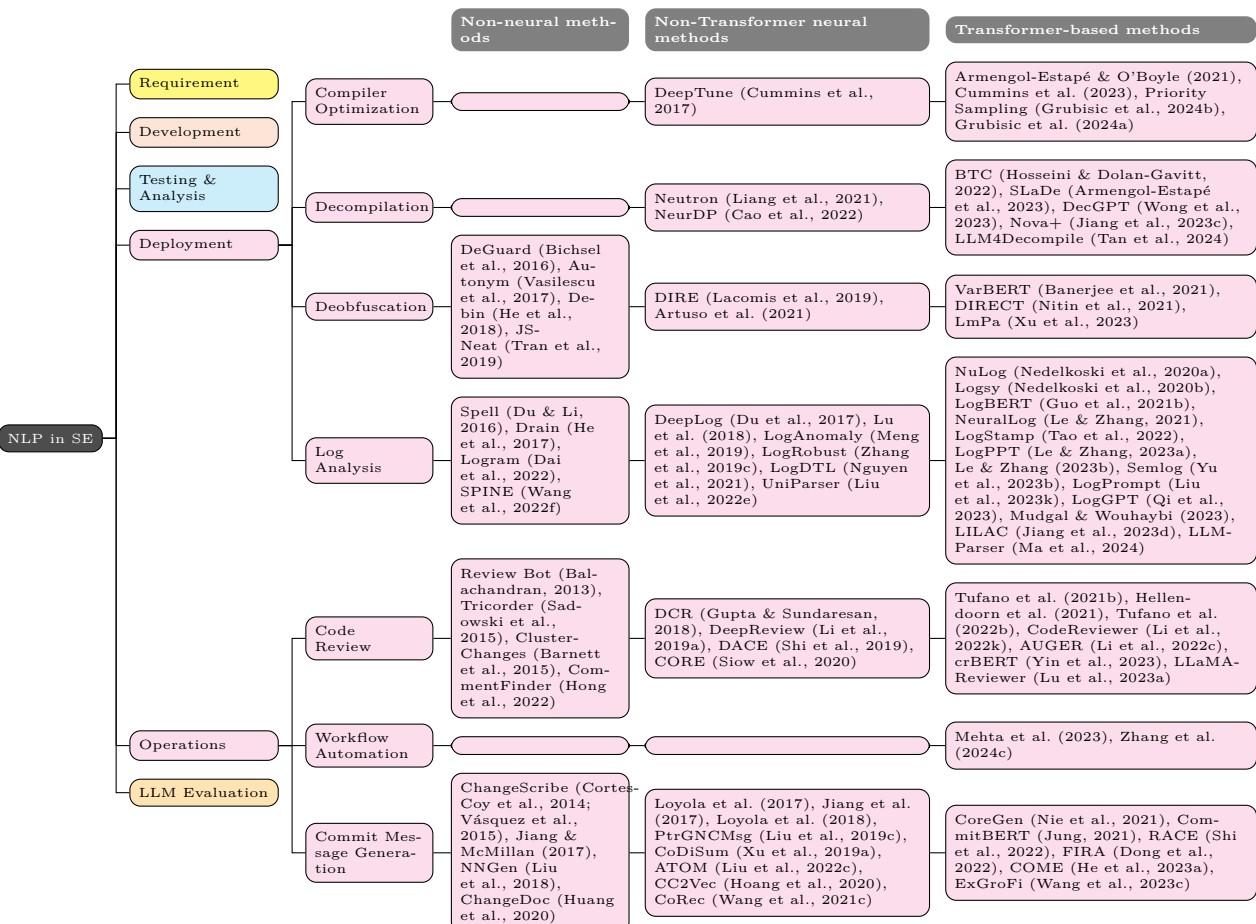

Figure 7: NLP applications in the *deployment & operations* stage of SE.

*Compiler optimization* is the task of optimizing compiled code to increase its efficiency. Many early works applied reinforcement learning and other machine learning technologies to this task by optimizing task scheduling or pass ordering (Wang & O'Boyle, 2018; Leather & Cummins, 2020), and it is only recently that researchers started to view it as a sequence-to-sequence generation task with the help of powerful LLMs (Cummins et al., 2023).

- *Decompilation* is the reverse process of compilation, and an important topic in reverse engineering. In this task a model takes a compiled program - represented in assembly code or binary machine code - and aims to output the original source program in high-level languages such as C.

- *Obfuscation* refers to the process of renaming identifiers (e.g. variables, methods, and classes), for example to generic names like `var_1, var_2` or `x, y`. It is an important technique in virus detection, intellectual property protection, and code size reduction (Collberg & Thomborson, 2002; Murad et al., 2010; Vasilescu et al., 2017). *Deobfuscation* refers to the reverse process, where meaningful identifier names are recovered from obfuscated programs. Obfuscation can be easily achieved statically, but deobfuscation has been a subject of more interest in recent years. It plays a significant role in decompiling, and has also been adopted as a pretraining objective for code language models (Lachaux et al., 2021; Ding et al., 2022a; Liu et al., 2022d).

- *Log analysis* aims to analyze the system logs produced by software products, for example parsing logs into structured templates or finding anomalies from raw logs. Zhu et al. (2019) provide a survey on traditional methods for this task up to 2018, and Chen et al. (2021d) give an empirical comparison between neural

network based methods. Zhang et al. (2023e) also cover more recent methods for log parsing, while Landauer et al. (2022) survey methods for anomaly detection in logs.

### Operations

- *Code review* aims to automate the process of peer code review, and includes many subtasks, such as review necessity prediction, review comment generation, code refinement, and review decision prediction.

- *Commit message generation* aims to automatically generate commit messages for code changes. This task takes the code before and after change as input, and output the description for the change. This can be viewed as the dual task of program repair, as many code changes and their accompanying commit messages concern bug fixing. Tao et al. (2021) provide a survey on methods and datasets for this task up to 2021.

- *Workflow Automation* is the automation of workflows in Git. Due to the complexity involved in this task, it was only recently brought to attention after the advent of powerful LLMs.

### 2.1.5   LLM Evaluation

Apart from the previously covered stages in SE, we observe that programming-related evaluation of LLMs has been gaining momentum in both the NLP and the SE community since the release of Codex. Thus, we also list several novel tasks that appeared recently for this purpose. We discuss this topic in more detail in Section 7.

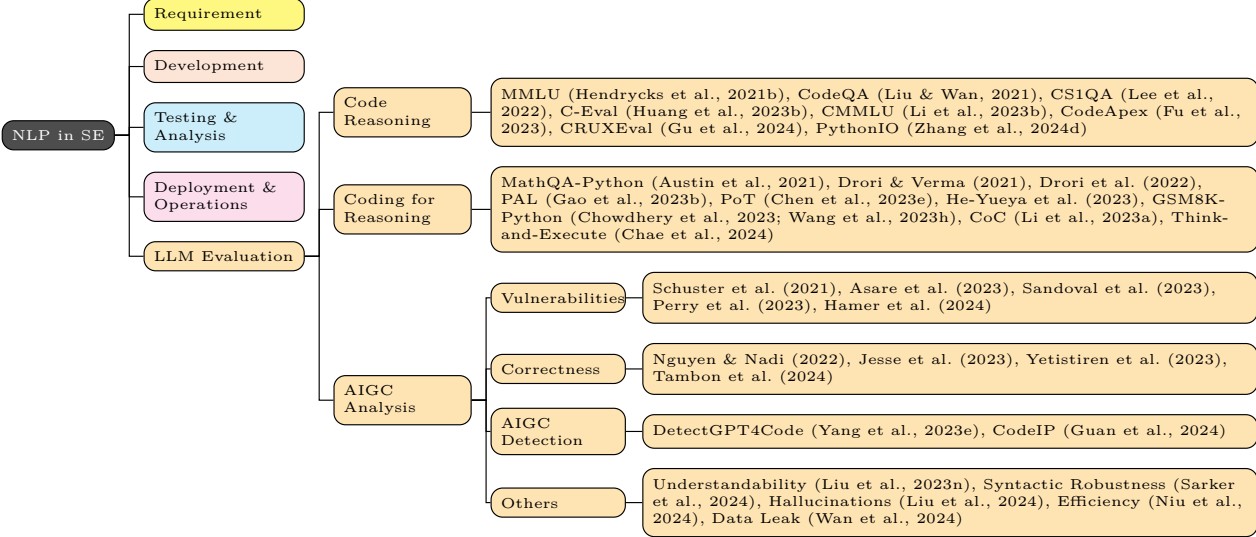

Figure 8: Programming-related evaluation of LLMs.

- *Code reasoning* requires a model to reason about code or algorithms, and answer related questions which are written in multiple-choice format or free-form QA format, which may range from conceptual understanding to numerical calculation and complexity analysis. It often comes as a subset of composite evaluation benchmarks such as MMLU (Hendrycks et al., 2021b).

- *Coding for reasoning* is a special case of reasoning where LLMs solve complex reasoning problems by generating code that will be executed by external interpreters. This task abstracts the reasoning process from numerical calculations, and is thus of special interest in evaluating LLMs. More recently research has also shown that asking the LLM itself to simulate the execution of generated code can also help its reasoning process.

- *AIGC analysis* aims to analyze AI-generated code[2] from various aspects, such as correctness, vulnerabilities, and detection of AI-generated code.

### 2.1.6 NLP Point-of-View

Unlike software engineering, evaluation tasks in NLP are generally categorized into understanding and generation. The former, represented by GLUE (Wang et al., 2018a) and SuperGLUE (Wang et al., 2019), emphasizes the comprehension of input text, and is typically formalized as classification, regression, sequence tagging, or span extraction. The later, on the other hand, involves autoregressive generation of text, such as machine translation and summarization.

Among the previously listed tasks, code synthesis, code translation, code repair, deobfuscation, unit test generation, assertion generation, mutant generation, code summarization, code review, identifier prediction, and commit message geneartion are sequence-to-sequence generation tasks. Formally, each instance of these tasks has a source sequence $\mathbf{x}$ (e.g. a piece of source code) and a target sequence $\mathbf{y}$ (e.g. its corresponding summarization), and the language model is tasked to maximize the conditional probability given by (5), where $\theta$ can be either a decoder-only model or an encoder-decoder model. In the former case, $\mathbf{x}$ and $\mathbf{y}$ are concatenated. In the later case, $\mathbf{x}$ is processed by the encoder and $\mathbf{y}$ is processed by the decoder.

Code completion and code infilling are also generation tasks, but differ from sequence-to-sequence tasks where the input and output are related by different sequences. In these two tasks, the target is a continuation or infill of the input. They correlate closely to the language modeling objectives given in Equation (3) and (5). Similarly, cloze test takes the same form as Equation (4) but is usually considered an understanding task, as its output is usually a single token and does not involve autoregressive generation.

Defect detection, malware detection, clone detection, code classification, and author identification are sequence classification tasks. In these tasks, a set of labels $\mathcal{Y}$ is defined over the input, and each instance is assigned a label $y \in \mathcal{Y}$ (e.g. for defect detection $\mathcal{Y} = \{0, 1\}$, while for author identification a possible $\mathcal{Y}$ is {Alice, Bob, John, others}). The model is then tasked to maximize

$$p_\theta(y|\mathbf{x}). \tag{1}$$

Type prediction is a token classification task, also known as tagging. In this task, each token $x_i$ is assigned a label $y_i \in \mathcal{Y}$, with an example $\mathcal{Y}$ being {int, float, string, bool, non-identifier, other}. The model's objective is to maximize

$$\prod_{i=1}^{n} p_\theta(y_i|\mathbf{x}). \tag{2}$$

The last two tasks - code retrieval and code search - also belong to understanding tasks. In these tasks, each source sequence $\mathbf{x}$ is paired with a positive target sequence $\mathbf{y}$ and a set of negative targets $\bar{\mathbf{y}} \in \{\mathbf{y}_1, \cdots, \mathbf{y}_k\}$. The model's task is to find a similarity metric $s$ such that $s(\mathbf{x}, \mathbf{y})$ is larger than $s(\mathbf{x}, \bar{\mathbf{y}})$.

### 2.1.7 Code LLMs for Low-Resource, Low-Level, and Domain-Specific Languages

In NLP, human languages are categorized into high-, middle-, and low-resource languages based on the amount of available data in each language. High-resource languages such as English are extensively studied, while low-resource languages such as Swahili and Yoruba often rely on transfer learning from other languages to improve performance due to data scarcity (Conneau et al., 2020; Xue et al., 2021; Zhang et al., 2023g).

This phenomenon also exists in code modeling, as most works listed in Figure 3-8 target the most popular programming languages such as C, Java, and Python. However, a major difference between NLP and SE is that in NLP, low-resource languages are often spoken by few people or even endangered, while in SE low-resource languages are often designed for specific domains and purposes, and thus have an active user community. Verilog, a hardware description language, is one such example, for which code modeling research

---

[2]In the AI community "AIGC" usually refers to AI-Generated *Content*. In this work we overload this expression with AI-Generated *Code*.

is quite abundant (Thakur et al., 2023a;b; Lu et al., 2023b; Liu et al., 2023g; Tsai et al., 2023; Thorat et al., 2023; Liu et al., 2023h; Nadimi & Zheng, 2024).

From a different perspective, most of these popular languages are imperative languages, while few works have studied NLP and LLM applications in declarative or functional languages, except those concerning SQL and one recent work on Haskell (van Dam et al., 2024). However, research has shown that declarative languages are more aligned for optimization since they describe the "what" and not "how" in programming (Marcus et al., 2019; 2020), thus they are worth more attention in the future research.

Similar to NLP, many works in code modeling have studied the transfer of model capability across languages. Chen et al. (2022), for example, investigate the performance of multilingual language models on Ruby (which is one of the six languages in the popular CodeSearchNet pertaining corpus (Husain et al., 2019), but relatively low-resourced compared with the other five), finding that multilingual models have lower performance-to-time ratio compared with monolingual ones. Cassano et al. (2023a), on the other hand, propose MutiPL-T, an approach for translating high-resource training data into low-resource languages for fine-tuning. However, such research is yet scarce compared with the NLP community, but with the recent advent of colossal, multilingual datasets such as The Stack v2 (Lozhkov et al., 2024) we expect to see more of it in the future.

## 2.2 Evaluation Metrics

Of the tasks mentioned in Section 2.1, the understanding tasks are similar in form to natural language understanding tasks (Wang et al., 2018a; 2019) and evaluated likewise by metrics such as accuracy, F1 and Mean Reciprocal Rank (MRR), while short generation tasks such as identifier prediction is also evaluated by accuracy of exact matches. Code-to-text tasks are evaluated with common metrics for text generation such as BLEU (Papineni et al., 2002).

Evaluation of tasks involving code generation, on the other hand, is more complicated. Most early works evaluate syntactical correctness, i.e. the percentage of generations that can be successfully parsed. Chen et al. (2018) argue against such metrics and suggest reference match instead, which is the percentage of generations that are exactly the same as the references. Ren et al. (2020) propose CodeBLUE, a variant of BLEU that takes code syntax and semantics into account by evaluating the overlap of abstract syntax tree (AST) and data flow.

As code generation models became more capable over the years, however, these metrics based on content-overlap have been found to be inadequate (Rozière et al., 2020; Hendrycks et al., 2021a; Austin et al., 2021), since functionally equivalent snippets of code can differ dramatically in their lexical forms. Consequently, researchers have turned their attention to functional correctness. One popular example of such metrics is pass@$k$, proposed by Kulal et al. (2019) and refined by Chen et al. (2021b), which is an unbiased estimator of the model's chance in passing all unit tests of a program with any of $k$ generated samples. This metric can be generalized to pass$n$@$k$ (Li et al., 2022h), which limits the number of model submissions to $n$ but allows filtering by unit tests given in the input from $k$ samples.

## 2.3 Program Synthesis

While dozens of evaluation tasks exist in software engineering, they have generally stayed out of the focus of the NLP community until very recently. The only exception is program synthesis, which has become a standard evaluation task for LLMs since the advent of HumanEval in 2021. Looking back at this task, we identify four changes in program synthesis over the years: shift of coding paradigms (from example-based to intention-based), generalization in languages (from domain-specific languages to general-purpose languages), simplification of model architectures (from grammar-guided decoders to general-purpose language models), and application of execution-based feedback.

Many of the early methods for program synthesis are example-based (Menon et al., 2013), which means they induce programs from input-output examples, often in domain-specific languages (DSLs) such as Flash-

Fill (Devlin et al., 2017a) and Karel[3] (Bunel et al., 2018), as these languages are usually simple in syntax and structure.

As code generation models became more capable over the years, researchers started to pay attention to program synthesis in general-purpose programming languages as well. Hearthstone (Ling et al., 2016) and CONCODE (Iyer et al., 2018) are two of the early datasets, representing Python and Java respectively. Each example in Hearthstone is the description of a card in the game and its corresponding class implementation, while examples in CONCODE are simply Java methods paired with their natural-language documentation crawled from public GitHub repositories. Synthesizing programs from their corresponding natural language descriptions has since then become a standard practice in program synthesis, and has led to some of the most widely used benchmarks, such as HumanEval (Chen et al., 2021b), which has even been translated into multiple languages (Cassano et al., 2023b; Zheng et al., 2023a; Muennighoff et al., 2024). Some recent benchmarks use general-purpose languages but focus on specific domains, such as data science (Bavishi et al., 2019; Lai et al., 2023) or Jupyter notebooks (Agashe et al., 2019), while several math reasoning benchmarks have also been converted to programming tasks, including MathQA-Python (Amini et al., 2019; Austin et al., 2021) and GSM8K-Python (Cobbe et al., 2021; Chowdhery et al., 2023; Wang et al., 2023h).

Many early works argue that simply treating program synthesis as a text generation task does not utilize the underlying syntax of programming languages, and thus often use syntax-enhanced decoders to inject the target syntax as prior knowledge (Yin & Neubig, 2017). LLMs, however, have demonstrated that pretrained language models are capable of generating syntactically correct programs without loss of generality. Under this setting, researchers start to *execute* the generated programs and provide feedback to the generation model to inject the prior knowledge of code instead. This has recently led to the popularity of *interactive coding*, which we discuss in more detail in Section 7.1.

### 2.4 Repository-Level Coding

Most tasks discussed in Section 2.1 are limited to a single file or even a single function, as cross-file code modeling poses challenges that are beyond the capability of most existing language models. Recently, however, position interpolation techniques (Chen et al., 2023d; Rozière et al., 2023; Peng et al., 2023a) have extended the context window of LLMs to hundreds of thousands of tokens, making it possible to contextualize coding activities within entire repositories. Several works have studied code completion (Shrivastava et al., 2023b; Ding et al., 2022b; Zhang et al., 2023b; Shrivastava et al., 2023a; Phan et al., 2024; Wu et al., 2024b) and generation (Liao et al., 2023; Zan et al., 2024) leveraging repository-level context, and corresponding benchmarks have been proposed Liu et al. (2023j); Ding et al. (2023); Li et al. (2024a). Bairi et al. (2023) investigate the more challenging tasks of repository-level API migration and temporal editing, while Jimenez et al. (2023) introduce a related benchmark, SWE-bench.

## 3 Language Modeling Preliminaries

As code is ultimately a subset of natural languages, language models have been extensively used to tackle the tasks listed in Section 2. Before diving into the language models themselves, we first briefly review the preliminaries of Transformer-based language modeling in this section following the common choices of training objectives, and also some implementation designs.

### 3.1 Causal Language Modeling

Unidirectional language models (also known as causal language models[4]) factor the probability of a sentence into the product of each token's conditional probability with the chain rule. A piece of input text $\mathbf{x} =$

---

[3]FlashFill is used in Microsoft Excel for string transformation. Karel is a simple programming language for educational purpose.

[4]The training objective of such language models is Causal Language Modeling (CLM), but also referred to as Next Token Prediction.

$[x_1, x_2, \cdots, x_n]$ consisting of $n$ tokens is modeled as

$$P(\mathbf{x}) = \prod_{i=1}^{n} p_\theta(x_i|\mathbf{x}_{1:i-1}), \tag{3}$$

where $\mathbf{x}_{1:i-1}$ is a shorthand for tokens before $x_i$ in the input, and $\theta$ is the parameters of the model. With Transformer decoders such as GPT (Radford et al., 2018; 2019; Brown et al., 2020) and LLaMA (Touvron et al., 2023a;b), the conditional probability in (3) is modeled by adding an attention mask to the attention matrix of each Transformer block, ensuring that $x_i$ can only attend to previous tokens. During training, the cross entropy loss on all tokens in the input is calculated in parallel, while at inference time each new token is generated autoregressively. For further details about the Transformer architecture we refer to Vaswani et al. (2017).

### 3.2   Masked Language Modeling

Unlike causal language models, bidirectional language models are trained to acquire a better contextual representation of text rather than generating text autoregressively. In the vanilla Transformer, the encoder part is allowed to attend to a token's left as well as right context for this purpose. BERT (Devlin et al., 2019) takes one step further and pretrains only a Transformer encoder. A set $\mathcal{M}$ of randomly chosen tokens in the input are replaced by a special token [MASK] to obtain a noisy input $\hat{\mathbf{x}}$, for example $[[\text{CLS}], x_1, [\text{MASK}], x_3, [\text{MASK}], x_5, [\text{EOS}]]^5$, and the model is trained to recover the original tokens by maximizing

$$\prod_{m \in \mathcal{M}} p_\theta(m|\hat{\mathbf{x}}). \tag{4}$$

While this objective requires the model to have a deep understanding of the input text to reconstruct it, it suffers from low training efficiency, since only a small set of tokens (usually 15%) are masked (and thus "trained on"). To address this issue, Clark et al. (2020) propose ELECTRA, which is trained to discriminate whether or not each token in the input has been replaced by a BERT-like model instead, thereby computing loss on all input tokens.

### 3.3   Denoising Objectives

GPT-style causal LM and BERT-style bidirectional LM each has its own strengths and weaknesses. While GPT can be used for autoregressive generation, it lacks a bidirectional representation of input text, and is thus unsuitable for sequence-to-sequence (seq2seq) generation tasks such as translation and summarization. BERT, on the other hand, can produce bidirectional representations, but is pretrained only for mask filling, not generation.

The vanilla Transformer encoder-decoder architecture combines the respective advantages of GPT and BERT. T5 (Raffel et al., 2020) is such a model pretrained with *span corruption*, which can be regarded as a variant of MLM. During pretraining, spans of text in the input are replaced with sentinel tokens, which play the same role as [MASK] in BERT. The noisy input is first processed by the encoder with bidirectional attention, and the masked spans are then generated autoregressively by the decoder. Formally, if $k$ spans are sampled for corruption in input $\mathbf{x}$, the noisy input $\hat{\mathbf{x}}$ is then constructed by replacing each span with a special token <extra_id_i>, for $i = 1, 2, \cdots, k$, and the target $\mathbf{y}$ is constructed by concatenating all spans prepended with corresponding sentinels: $[\text{<extra\_id\_1>}, \text{span}_1, \cdots, \text{<extra\_id\_k>}, \text{span}_k]$. The model is then trained with a standard seq2seq objective, by maximizing

$$p_\theta(\mathbf{y}|\hat{\mathbf{x}}) = \prod_{i=1}^{n_y} p_\theta(y_i|\hat{\mathbf{x}}, \mathbf{y}_{1:i-1}). \tag{5}$$

---

[5]Both [CLS] and [EOS] are artificial tokens added to the input text. [CLS] is added at the beginning and its representation is used for sentence classification, while [EOS] indicates end of sentence. The original BERT also uses another special token [SEP], which is not in common use in LLMs, and we refer to Devlin et al. (2019) for details.

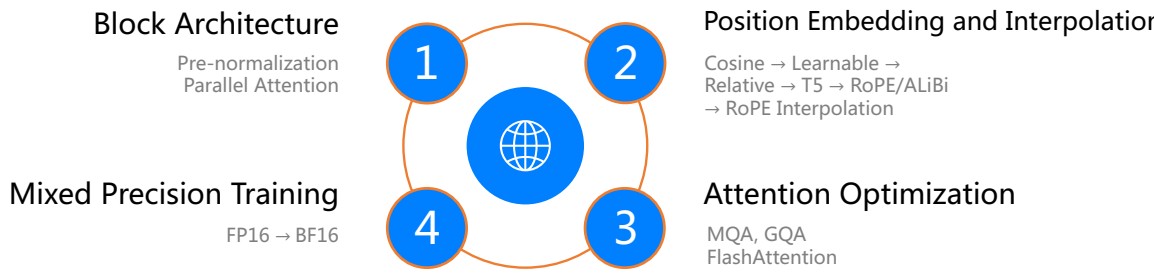

Figure 9: Major implementation changes in LLM over the past few years.

Lester et al. (2021) show that models pretrained with such objectives can be adapted for autoregressive language modeling with extra pretraining using the prefix language modeling objective, i.e. splitting the text into two parts, processing the first part with encoder and generating the second part with decoder.

Tay et al. (2023b) argue that span corruption is also closely related to CLM, since one can mask out the whole input text as a single span and train the decoder to generate it autoregressively. Inspired by this relation, they propose UL2, which is the combination of many span corruption objectives that differ in corruption rate and span length. Applying it to both encoder-decoder models and decoder-only models, they find that encoder-decoder models perform better under the same computation budget constraint. Other researches have also found that such encoder-decoder models generally perform better than causal decoder-only models (Wang et al., 2022c; Soltan et al., 2022).

### 3.4 Auxiliary Objectives

Language modeling objectives, such as previously discussed CLM and MLM, mainly train the model to capture token-level information and are ineffective at modeling document structures. Thus, auxiliary objectives are often added to help the models learn such global information. BERT is pretrained with next sentence prediction (NSP) along with MLM, which is formulated as a binary classification task to predict whether two segments in the input are neighboring in the original corpus. Lan et al. (2020) propose a more challenging sentence-order prediction (SOP) task, where the negative samples are constructed by swapping the order of two neighboring sentences instead of sampling a random sentence from other documents.

Relatedly, Raffel et al. (2020) mix supervised downstream samples such as GLUE (Wang et al., 2018a) into T5's pretraining dataset to conduct multi-task pretraining. However, it is worth noting that since they unify all tasks into a text-to-text format, the training objective is the same for their self-supervised pretraining and supervised downstream tasks, i.e. conditional generation of the target sequence.

### 3.5 Implementation Design

While most researches on pretraining language models have focused on designing novel training objectives, low-level implementation of the Transformer architecture itself is also being continuously improved over the years in pursuit of stability, performance, and efficiency, as shown in Figure 9.

The original Transformer block proposed by Vaswani et al. (2017) is formulated as

$$h = \text{LN}(\text{Attention}(x) + x), \tag{6}$$
$$y = \text{LN}(\text{FFN}(h) + h), \tag{7}$$

where $x$ is the layer's input; $y$ is the layer's output; "Attention" is the self-attention sublayer; "FFN" is the feed-forward sublayer, and "LN" is layer normalization (Ba et al., 2016).

GPT-2 (Radford et al., 2019) moves layer normalization to the input of each Transformer sub-block to stabilize training:

$$h = \text{Attention}(\text{LN}(x)) + x, \tag{8}$$
$$y = \text{FFN}(\text{LN}(h)) + h, \tag{9}$$

and such pre-norm has since become a standard practice in Transformer decoders.

GPT-J (Wang & Komatsuzaki, 2021) modifies the Transformer block to compute FFN sub-layer and self-attention sub-layer in parallel to increase computation throughput:

$$y = x + \text{FFN}(\text{LN}(x)) + \text{Attention}(\text{LN}(x)), \tag{10}$$

and Chowdhery et al. (2023) observes limited performance degradation when applying this design to larger models.

As self-attention alone cannot capture the position information of each input token, the vanilla Transformer adds a non-learnable vector that's dependent on the absolute position in the input sequence to the embedding of each token, which is known as cosine position embedding. Later works such as GPT and BERT use learnable embeddings instead, while T5 adopts relative position embedding (Shaw et al., 2018), where the information of relative position between query and key is injected in the computation of self-attention instead. A more recent scheme RoPE (Su et al., 2024) multiplies the keys and queries by a position-dependent rotation matrix, and is later shown to enable position interpolation for processing of longer sequences (Chen et al., 2023d; Rozière et al., 2023; Peng et al., 2023a). Alternatively, Press et al. (2022) propose ALiBi, which directly attenuates the attention scores according to the relative position between key and query. RoPE is popularized by its application in PaLM (Chowdhery et al., 2023) and GPT-NeoX (Black et al., 2022), while ALiBi is adopted by BLOOM (Scao et al., 2022). We refer to Zhao et al. (2023) for a survey on position embedding and interpolation in Transformers.

Apart from position embeddings, another issue in Transformer that has long troubled researchers is the fact that the complexity of self-attention scales quadratically with the input sequence length. Some works such as Reformer (Kitaev et al., 2020), Linformer (Wang et al., 2020c), Performer (Choromanski et al., 2021) and cosFormer (Qin et al., 2022b) use approximate attention to reduce this complexity, but they mostly come at the cost of degraded performance (Tay et al., 2023a; Lin et al., 2022). Other works tackle this issue from an engineering point-of-view. MQA (Shazeer, 2019) shares the same set of keys and values across all attention heads to optimize memory-to-arithmetic ratio and significantly improves inference speed at small costs of model performance. Its variant Grouped-Query Attention (GQA, Ainslie et al., 2023) takes a middle-ground approach by dividing attention heads into groups and sharing the same set of keys/values within each group. Orthogonally, Dao et al. (2022) introduce FlashAttention, which is an exact but improved implementation of self-attention that optimizes IO operations on the accelerating device via tiling to improve memory efficiency.

Another important technique for improving LLMs' training efficiency is mixed precision training (Micikevicius et al., 2018), which stores model weights and activations in lower precision to save memory consumption. Early works use FP16 format for this low precision, while BF16 has now become more popular (Chowdhery et al., 2023; Scao et al., 2022), as it can represent the same range of floating point numbers as FP32, thus preventing numerical overflowing and underflowing during training.

## 4 General Language Models for Code

Since language models scaled to hundreds of billions of parameters (Brown et al., 2020; Chowdhery et al., 2023), many of them have demonstrated non-trivial coding capability, even if they are not specifically designed or trained for code. Pioneered by Codex, researchers have also found continual pretraining on code to significantly benefit language models' performance on code[6].

---

[6]While some works refer to this process as "finetuning on code", it is still self-supervised in nature. Thus we choose to adopt the term "extra/additional/continual pretraining" in this work to avoid confusion with supervised in-task finetuning or instruction finetuning.

## 4.1 Off-the-Shelf Language Models

Large language models are often pretrained on trillions of tokens following the scaling laws (Kaplan et al., 2020; Hoffmann et al., 2022), and such an amount of text data is often a diverse composite with a non-negligible part of code. The Pile (Gao et al., 2021), for example, includes 95GB of code crawled from GitHub out of its 800GB raw dataset, while the multilingual pretraining dataset ROOTS (Laurençon et al., 2022) also contains 163GB of code spanning 13 programming languages in its 1.6TB compound. As two of the largest open-source pretraining datasets, they have supported many language models with coding ability. GPT-J (Wang & Komatsuzaki, 2021), for example, is reported by Chen et al. (2021b) to demonstrate non-trivial performance on HumanEval, while Scao et al. (2022) report similar results for GPT-NeoX (Black et al., 2022) and BLOOM. LLaMA (Touvron et al., 2023a), whose pretraining dataset includes 328GB code from GitHub, achieves 23.7 pass@1 performance on HumanEval, and its successor LLaMA 2 (Touvron et al., 2023b), achieves an even higher score of 29.9.

Closed-source models, on the other hand, perform generally better. LaMDA (Thoppilan et al., 2022) and PaLM (Chowdhery et al., 2023), whose pretraining dataset contains 12.5% and 5% code respectively, achieve 14.0 and 26.2 pass@1 performance on HumanEval, while GPT-4 (OpenAI, 2023) set a staggering record of 67.0 (and an early version is reported by Bubeck et al. (2023) to be 82) that until recently has remained higher than any specialized models pretrained or instruction-finetuned for code.

More recently, the general trend has been to train smaller models with larger datasets, following the revised scaling law (Hoffmann et al., 2022). Baichuan 2 (Yang et al., 2023a), for example, is a 13B model

Table 1: Pass@$k$ performance of raw language models (top) and language models with extra training on code (bottom) on HumanEval (0-shot) and MBPP (3-shot), ordered chronologically. For Phi-1.5 we consider Phi-1.5-web version, and for Code LLaMA we consider its Python version. [1] Chen et al. (2021b); [2] Chowdhery et al. (2023); [3] Austin et al. (2021); [4] Scao et al. (2022); [5] Touvron et al. (2023a); [6] OpenAI (2023); [7] Bubeck et al. (2023); [8] Touvron et al. (2023b); [9] Li et al. (2023j); [10] Yang et al. (2023a); [11] Bai et al. (2023); [12] Jiang et al. (2023a); [13] Anil et al. (2023a); [14] DeepSeek-AI et al. (2024); [15] Jiang et al. (2024a); [16] Dai et al. (2024a); [17] Anil et al. (2023b); [18] Rozière et al. (2023).

| | Size | HumanEval (0) | | MBPP (3) | |
|---|---|---|---|---|---|
| | | k=1 | k=100 | k=1 | k=80 |
| GPT-J[1] | 6B | 11.6 | 27.7 | | |
| LaMDA[23] | 137B | 14.0 | 47.3 | 14.8 | 62.4 |
| PaLM[2] | 540B | 26.2 | 76.2 | 36.8 | 75.0 |
| GPT-NeoX[4] | 20B | 15.4 | 41.2 | | |
| BLOOM[4] | 176B | 15.5 | 55.5 | | |
| LLaMA[5] | 65B | 23.7 | 79.3 | 37.7 | 76.8 |
| GPT-4 | | 67.0[6]/82[7] | | | |
| LLaMA 2[8] | 70B | 29.9 | 89.0 | 45.0 | 81.5 |
| Phi-1.5[9] | 1.3B | 41.4 | | 43.5 | |
| Baichuan 2[10] | 13B | 17.1 | | 30.2 | |
| Qwen[11] | 14B | 32.3 | | 40.8 | |
| Mistral[12] | 7B | 30.5 | | 47.5 | |
| Gemini[13] | Ultra | 74.7 | | | |
| DeepSeek[14] | 67B | 42.7 | | 57.4 | |
| Mixtral[15] | 8x7B | 40.2 | | 60.7 | |
| DeepSeekMoE[16] | 16B | 26.8 | | 39.2 | |
| Codex[1] | 12B | 28.8 | 72.3 | | |
| PaLM-Coder[2] | 540B | 36.0 | 88.4 | 47.0 | 80.8 |
| PaLM 2*[17] | S | 37.6 | 88.4 | 50.0 | 86.8 |
| Code LLaMA[18] | 34B | 53.7 | 94.7 | 56.2 | |
| Code-Qwen[11] | 14B | 45.1 | | 51.4 | |

trained on 2.6T tokens, while Qwen (Bai et al., 2023) is a 14B model trained on 3T tokens. They achieve 17.1 and 32.3 pass@1 on HumanEval, respectively. Li et al. (2023j), however, demonstrate that models as small as 1.3B can acquire coding capability that's comparable to much larger models while also maintaining a reasonable performance on general text processing and even manifesting some emergent abilities (Wei et al., 2022c) such as chain-of-though reasoning (Wei et al., 2022d). Their model, Phi-1.5, is trained on 21B tokens of textbook data generated by ChatGPT, and 100B tokens of filtered web data from Stack Overflow and Refined Web (Penedo et al., 2023), and attains 41.4 pass@1 performance on HumanEval.

Another rising trend is mixture-of-expert models (Lepikhin et al., 2021; Fedus et al., 2022; Du et al., 2022). Notably, Jiang et al. (2024a) recently introduce Mixtral 8x7B, where only 13B parameters are activated for each token during inference, but achieving 40.2 on HumanEval and surpassing much larger dense models such as LLaMA 2. Similarly, Dai et al. (2024a) present DeepSeekMoE, where only 2.8B parameters are activated in a 16B model, scoring 26.8 on HumanEval.

The exact performance of these models are presented in Table 1.

## 4.2 Language Models with Additional Pretraining on Code

Along with the seminal benchmark HumanEval, Chen et al. (2021b) kick-started the age of LLM for code with Codex, which are GPT-3 checkpoints pretrained on 100B additional code tokens and one of the earliest multi-billion models for code. Following their work, other researchers have also specialized their LLMs on code with additional pretraining. Chowdhery et al. (2023) train PaLM on 7.8B additional code tokens to obtain PaLM-Coder, setting new state-of-the-art on HumanEval and MBPP (Table 1) that are only broken later by its successor PaLM 2-S*, the smallest version of PaLM 2 (Anil et al., 2023b) further trained on an undisclosed amount of code. Similarly, Lewkowycz et al. (2022) train PaLM on 38.5B tokens of arXiv papers and mathematical content, while Rozière et al. (2023) train LLaMA 2 (Touvron et al., 2023b) on more than 500B code tokens to acquire Code LLaMA, whose performance on HumanEval surpasses all previous LMs except GPT-4 (Table 1). Liu et al. (2023b) further train Code LLaMA with multi-task finetuning (MFT) to introduce CodeFuse-CodeLLaMA, achieving 74.4 pass@1 on HumanEval and surpassing even the performance of GPT-4 published in OpenAI (2023).

While almost all of these models are Transformer decoders pretrained with CLM, several architectural modifications have been introduced along the way, as we noted in Section 3.5. All these models use pre-norm, and GPT-J introduces parallel attention, which is later adopted by PaLM, GPT-NeoX, and Phi-1.5. PaLM introduces MQA and RoPE into LLMs, and RoPE is now employed by most language models, including GPT-NeoX, two generations of LLaMA, Qwen, and the 7B version of Baichuan 2. BLOOM and the 13B version of Baichuan 2, however, use ALiBi for position embeddings, while LLaMA 2 and Code LLaMA adopt GQA instead of MHA or MQA. In Section 5, we show that specialized models pretrained exclusively on code have also followed these advancements closely.

# 5 Specialized Language Models for Code

As pretrained Transformers such as GPT and BERT achieved remarkable success in natural language processing, such model architectures, learning paradigms, and training objectives were soon adopted by the software engineering community to produce specialized models for code understanding and generation. In this section, we first review common datasets used for pretraining code language models (Section 5.1), and then dive into the complex family of code LMs by their model architecture: encoder-only models (Section 5.2), encoder-decoder models (Section 5.3), decoder-only models (Section 5.4), UniLM (Section 5.5), and diffusion models (Section 5.6). Lastly, in Section 5.7 we also illustrate the current trend of applying more recent techniques in NLP, such as instruction tuning (Wei et al., 2022b; Sanh et al., 2022; Chung et al., 2022) and reinforcement learning (Ouyang et al., 2022) to code processing. An overview of these pretrained models are provided in Table 3.

## 5.1 Training Dataset for Code

While text data for pretraining language models are often crawled from the web and must undergo meticulous and often aggressive preprocessing (Raffel et al., 2020), code data come naturally as whole documents from public GitHub repositories. Even better, they come with readily available quality indicators such as the count of stars or forks (although Allal et al. (2023) suggest that star count correlates poorly with downstream performance). As a result, many large-scale code pretraining datasets have been introduced, including CodeSearchNet (Husain

Table 2: Statistics of several pretraining datasets for code models: size in bytes, number of files, and number of programming languages. In CodeSearchNet each file is a function. For Pile and ROOTS we only consider their code composite. [a] Husain et al. (2019); [b] Gao et al. (2021); [c] Biderman et al. (2022); [d] https://huggingface.co/datasets/codeparrot/github-code; [e] Kocetkov et al. (2023); [f] Laurençon et al. (2022); [g] Lozhkov et al. (2024).

| Dataset | Size (GB) | Files (M) | # PL |
|---|---|---|---|
| CodeSearchNet[a] | 20 | 6.5 | 6 |
| The Pile[bc] | 95 | 19 | - |
| CodeParrot[d] | 1K | 115 | 30 |
| The Stack[e] | 3136 | 317 | 30 |
| ROOTS[f] | 163 | 15 | 13 |
| The Stack v2[g] | 32K | 3K | 619 |

et al., 2019), CodeParrot (Tunstall et al., 2022), and the Stack (Kocetkov et al., 2023), totaling 20GB, 50GB and 3TB of code documents respectively (Table 2).

While these datasets are meant for training code models, it should be noted that code is ultimately a special form of natural language, as the vocabulary of most programming languages is a small subset of English. Besides, high-quality code is often interleaved with natural language comments or documentations, which also enables models to acquire certain knowledge of general text representation. In fact, of the 6.5M functions in CodeSearchNet, 2.3M are paired with natural language documentation, allowing models to train explicitly on such bimodal data.

Compared with natural language, another byproduct of scraping code from GitHub is commit histories, which consist of code before commit, code after commit, and a short message describing the commit, which can loosely serve as an instruction for language models. Muennighoff et al. (2024) utilize this feature and construct a 2GB dataset CommitPackFT containing 742K samples of instruction data for code, obviating the need of extensive human labor that's required to construct natural language instructions (Sanh et al., 2022; Wang et al., 2022g).

Apart from bimodal training and instruction finetuning, another recent trend in constructing code dataset is synthesizing data with powerful models such as ChatGPT. While this method is originally proposed for generating instruction data in natural language (Wang et al., 2023g; Honovich et al., 2023), Gunasekar et al. (2023) take one step further and synthesize 1B tokens of Python textbooks and coding exercises to pretrain a 1.3B model, achieving state-of-the-art results on HumanEval that's comparable to much larger models trained on significantly larger datasets.

Table 3: An overview of pretrained code language models' architecture and training details: their base architecture, model size, vocabulary, context length, position embedding, training precision, attention type (MHA, MQA, or GQA), layer normalization type (post-norm or pre-norm), usage of FlashAttention, training initialization, objectives, dataset size (either in disk size, measured by GB/TB, or in number of tokens, measured by B/T), tokens seen during training, supported number of programming languages, and institute. We note that the number of training tokens does not count the training tokens of the model used for initialization, if any. The training objectives are: MLM (Masked Language Modeling), NSP (Next Sentence Prediction), RTD (Replaced Token Detection), IP (Identifier Prediction), CL (Contrastive Learning), SC (Span Corruption), DAE (Denoising Auto-Encoding), Text↔Code (text-to-code generation and code-to-text generation), MT (Machine Translation). Missing information (such as AlphaCode's position embedding type) is left as blank.

| Date | Model | Arch. | Size | Vocab | Context | PE | Precision | Atten. Type | Parallel Atten. | Pre-Norm | Flash Atten. | Init. from | Objectives | Dataset | Training | PL | Inst. |
|---|---|---|---|---|---|---|---|---|---|---|---|---|---|---|---|---|---|
| 2019-12 | CuBERT | BERT | 350M | 50K | 1024 | absolute | fp32 | MHA | | | | - | MLM + NSP | 9.3B | 93B | 1 | Google |
| 2020-02 | CodeBERT | RoBERTa | 125M | 50K | 512 | absolute | fp32 | MHA | | | | RoBERTa | MLM + RTD | 20GB | 105B | 6 | Microsoft |
| 2020-09 | GraphCode-BERT | RoBERTa | 125M | 50K | 640 | absolute | fp32 | MHA | | | | CodeBERT | MLM + Edge Prediction + Node Alignment | 20GB | 131B | 6 | Microsoft |
| 2021-08 | SynCoBERT | RoBERTa | 125M | 50K | 512 | absolute | fp32 | MHA | | | | CodeBERT | MLM + IP + AST Edge Prediction + CL | 20GB | 7B | 6 | Huawei |
| 2021-10 | DISCO | BERT | 100M | 20K | 512 | absolute | fp32 | MHA | | | | - | MLM + Node Type MLM + CL | 1.8GB | | 2 | Columbia & IBM |
| 2022-05 | Code-MVP | RoBERTa | 125M | 50K | 512 | absolute | fp32 | MHA | | | | GraphCodeBERT | MLM + Type Inference + CL | 2GB | 39B | 1 | Huawei |
| 2022-10 | SCodeR | RoBERTa | 125M | 51K | 1024 | absolute | fp32 | MHA | | | | UniXcoder | CL | 20GB | | 6 | Microsoft |
| 2020-05 | GPT-C | GPT-2 | 374M | 60K | 1024 | absolute | | MHA | | ✓ | | - | CLM | 11B | 270B | 4 | Microsoft |
| 2021-02 | CodeGPT | GPT-2 | 124M | 50K | 1024 | absolute | fp32 | MHA | | ✓ | | GPT-2 | CLM | 2GB | | 1 | Microsoft |
| 2022-02 | PolyCoder | GPT-NeoX | 160M-2.7B | 50K | 2048 | RoPE | fp32 | MHA | | ✓ | | - | CLM | 254GB | 39B | 12 | CMU |
| 2022-03 | CodeGen-Multi(Mono) | GPT-J | 350M-16.1B | 50K | 2048 | RoPE | fp16 | MHA | ✓ | ✓ | | - | CLM | 1.6TB(1.8TB)/ 506B(577B) | 1T(1.2T) | 6(1) | Salesforce |
| 2022-04 | InCoder | GPT-3 | 6.7B | 50K | 2048 | Cosine | fp32 | MHA | | ✓ | | - | Causal Masking | 204GB | 52B | 28 | Meta |
| 2022-06 | PyCodeGPT | GPT-Neo | 110M | 32K | 1024 | absolute | fp32 | MHA | | ✓ | | - | CLM | 96GB | 100B | 1 | Microsoft |
| 2022-07 | PanGu-Coder | PanGu-α | 317M-2.6B | 42K | 1024 | absolute | | MHA | | ✓ | | - | CLM | 147GB | 230B | 1 | Huawei |
| 2023-01 | SantaCoder | GPT-2 | 1.1B | 49K | 2048 | absolute | fp32 | MQA | | ✓ | | - | FIM | 268GB | 236B | 3 | BigCode |
| 2023-03 | CodeGeeX | PanGu-α | 13B | 52K | 2048 | absolute | fp16 | MHA | | ✓ | | - | CLM | 158B | 850B | 23 | Tsinghua |
| 2023-05 | StarCoder | GPT-2 | 15.5B | 49K | 8192 | absolute | fp32 | MQA | | ✓ | ✓ | - | FIM | 815GB | 1T | 86 | BigCode |
| 2023-05 | Jam | GPT-2 | 350M | 50K | 256 | absolute | bf16 | MHA | | ✓ | | - | CLM | 36GB/20B | 9B | 1 | U. Notre Dame |
| 2023-06 | Phi-1 | GPT-J | 1.3B | 51K | 2048 | RoPE | fp16 | MHA | ✓ | ✓ | ✓ | - | CLM | 7B | 53B | 1 | Microsoft |
| 2023-10 | CodeFuse | GPT-J | 350M-13B | 101K | 4096 | RoPE | fp16 | MHA | ✓ | ✓ | ✓ | - | CLM | 1.6TB / 1T | | 40+ | Ant Group |
| 2023-10 | CodeShell | GPT-2 | 7B | 70K | 8192 | RoPE | bf16 | GQA | | ✓ | | - | FIM | 100B | 500B | | Peking U. |
| 2020-10 | PyMT5 | GPT-2 | 374M | 50K | 1024+1024 | absolute | fp16 | MHA | | ✓ | | - | SC | 27GB | | 1 | Microsoft |
| 2021-02 | Mastropaolo et al. | T5 | 60M | 32k | 512+512 | T5 | bf16 | MHA | | ✓ | | - | SC | 1GB | | 1 | USI |
| 2021-02 | DOBF | | 250M | 50K | 512+512 | absolute | fp16 | MHA | | | | - | MLM + Deobfuscation | 45GB | | 2 | Meta |
| 2021-03 | PLBART | BART | 140M | 50K | 1024+1024 | absolute | fp32 | MHA | | | | - | DAE | 655GB / 71B | 210B | 2 | UCLA & Columbia |
| 2021-09 | CodeT5 | T5 | 60M-220M | 32K | 512+256 | T5 | fp16 | MHA | | ✓ | | - | SC + IP + Masked IP + Text↔Code | ∼25GB | | 8 | Salesforce |
| 2022-01 | SPT-Code | BART | 262M | 80K | 512+512 | absolute | fp16 | MHA | | | | - | NSP + SC + Method Name Prediction | 20GB | | 6 | Nanjing U. |
| 2022-02 | AlphaCode | | 300M-41B | 8K | 1536+768 | | bf16 | MQA | | | | - | MLM + CLM | 715GB | 967B | 13 | DeepMind |
| 2022-06 | NatGen | T5 | 220M | 32K | 512+256 | T5 | fp16 | MHA | | ✓ | | CodeT5 | Naturalization | ∼26GB | 14B | 8 | Columbia & UCD |
| 2022-12 | ERNIE-Code | mT5 | 580M | 250K | 1024+1024 | T5 | bf16 | MHA | | ✓ | | mT5 | SC + Text↔Code + MT | | 197B | 6 | Baidu |
| 2023-05 | CodeT5+ | T5/GPT-3 | 220M-16B | 50K | 2048+2048 | absolute | fp16 | MHA | ✓ | ✓ | | CodeGen-mono | SC + CLM + CL + Text↔Code | 52B | | 9 | Salesforce |
| 2020-12 | CugLM | BERT | 51M | 50K | 128 | absolute | fp32 | MHA | | | | - | MLM + NSP + CLM | 8M | 1.2B | 2 | Peking U. |
| 2022-03 | UniXcoder | RoBERTa | 125M | 51K | 1024 | absolute | fp32 | MHA | | | | - | MLM + CLM + SC + CL + Code2Text | 20GB+ | 839B | 6 | Microsoft |

## 5.2 Encoders

Pretrained Transformer encoders such as BERT (Devlin et al., 2019), RoBERTa (Liu et al., 2019d), and ELECTRA (Clark et al., 2020) have attained impressive results on natural language understanding tasks, and these methods were soon introduced into code processing after their advent. Kanade et al. (2020) replicate the training procedure of BERT on a code corpus to produce CuBERT, showcasing its superior performance over LSTM (Hochreiter & Schmidhuber, 1997) and non-pretrained Transformers. Feng et al. (2020), on the other hand, train CodeBERT with MLM and ELECTRA's RTD on CodeSearchNet. They also utilize the explicit text-code pairs in CodeSearchNet, and use them respectively as the first and second segment in BERT's input. When using CodeBERT to initialize the encoder part of a vanilla Transformer for sequence-to-sequence generation tasks such as code summarization, they observe a moderate performance gain over non-pretrained baselines.

Apart from these standard training objectives, many auxiliary objectives specifically designed for code have also been introduced. GraphCodeBERT (Guo et al., 2021a) and SynCoBERT (Wang et al., 2021d) both extract graphs from the source code (data flow graph and abstract syntax tree, respectively) and train the models to predict the typological relations between the nodes, while SynCoBERT and Code-MVP (Wang et al., 2022e) also add type inference to their pretraining stage in the form of tagging. Another common objective is contrastive learning: SynCoBERT and Code-MVP contrast between different views of the input (such as code, comment, AST, and transformed code), while DISCO (Ding et al., 2022a) constructs positive sample pairs by semantic-preserving transformations such as obfuscation, and negative pairs by injecting artificial bugs.

## 5.3 Encoder-Decoders

In NLP, pretrained Transformer encoder-decoders such as T5 (Raffel et al., 2020) and BART (Lewis et al., 2020) have also left a notable mark in the past few years' advancement in language modeling. T5, for example, unifies all textual tasks into a sequence to sequence format and sets new records on GLUE (Wang et al., 2018a) and SuperGLUE (Wang et al., 2019). Compared with encoder-only models, encoder-decoders are naturally more powerful as they can be used for conditional text generation, while their encoder part can always be taken out to perform tasks that require an encoder-only architecture, such as regression (Tay et al., 2023b).

Inspired by these advantages of encoder-decoder architecture, many such models have been proposed for code processing. PyMT5 (Clement et al., 2020) and Mastropaolo et al. (2021) replicate the pretraining and multi-task finetuning process of T5 on code corpus, while Ahmad et al. (2021) introduce PLBART, a BART pretrained on 655GB combined data of Java, Pyhton, and natural language. Lachaux et al. (2021) argue that MLM could be too easy a task for programming languages as identifier names often occur multiple times in a single context window, and propose a deobfuscation pretraining objective, where the model is trained to convert obfuscated code back to its original form. Related to this method, we note that meaningful variable names have also been found to have a positive impact on the code generation process of large language models (Chen et al., 2023e).

Building on these early works, Wang et al. (2021f) propose CodeT5, which is pretrained alternatively with 1) T5's original span corruption; 2) identifier tagging (where each token in the code input is tagged as either identifier or non-identifier); 3) masked identifier prediction (a special form of span corruption where all identifiers are masked); and 4) text-to-code & code-to-text generation. Its successor, CodeT5+ (Wang et al., 2023h), take inspiration from UL2 (Tay et al., 2023b) and introduce causal language modeling (CLM) into pretraining, along with additional contrastive objectives based on text-code matching.

AlphaCode (Li et al., 2022h) is also trained with multiple objectives, where the encoder is trained with MLM and the decoder is trained with CLM, with architecture modifications such as shallow-encoder & deep-decoder, multi-query attention (Shazeer, 2019), and being much larger than CodeT5 (up to 41B parameters). NatGen (Chakraborty et al., 2022a), on the other hand, is pretrained with a "naturalization" objective similar to deobfuscation: semantically equivalent but unnatural code is generated by predefined operations such as loop transformation, dead code injection, and variable renaming, and the model is pretrained to translate

these unnatural code back to its original form. We note that some of these models are built on previous works. For example, NatGen is initialized with CodeT5, while the largest version of CodeT5+ is initialized from a decoder-only model, CodeGen (Nijkamp et al., 2023b).

Apart from these general pretraining objectives, several works have also trained Transformer encoder-decoders with a focus on code translation, which is a natural application of Transformer models in code as the Transformer architecture was originally proposed by Vaswani et al. (2017) for machine translation (MT). However, unlike natural languages, where parallel corpus across two or more human languages exist in abundance, there is little parallel data for code. To tackle this issue, Rozière et al. (2020) propose Transcoder, which first pretrains an encoder with XLM (Conneau & Lample, 2019), and then initializes a vanilla Transformer with this encoder and continue to pretrain it with Denoising Auto-Encoding (DAE, Lewis et al., 2020) and back translation (Sennrich et al., 2016), while its follow-up work (Szafraniec et al., 2023) also utilize language-independent intermediate representations to enhance this process, which we discuss in more detail in Section 6.

Apart from training data and objectives, these models mostly keep to the original architectures proposed by the NLP community, as shown in Table 3. Models based on BART, for example, use post-normalization and learnable absolute position embeddings, while those based on T5 use its simplified relative position embeddings and pre-normalization.

Table 4: Pass@1 performance of pretrained code models (top), instruction finetuned code models (middle), in comparison with some of the best general language models (bottom), with models in each category ordered chronologically. The sources of these figures can be found in Section 5.3, Section 5.4, and Table 1.

| Model | Size | HumanEval | MBPP |
|---|---|---|---|
| PolyCoder | 2.7B | 5.6 | - |
| CodeGen-Mono | 16.1B | 29.3 | 35.3 |
| InCoder | 6.7B | 15.2 | 19.4 |
| PyCodeGPT | 110M | 8.3 | - |
| Pangu-Coder | 2.6B | 23.8 | 23.0 |
| SantaCoder | 1.1B | 14.0 | 35.0 |
| CodeGeeX | 13B | 22.9 | 24.4 |
| StarCoder | 15.5B | 33.6 | 52.7 |
| CodeT5+ | 16B | 30.9 | - |
| Phi-1 | 1.3B | 50.6 | 55.5 |
| CodeFuse | 13B | 24.8 | - |
| DeepSeek Coder | 33B | 56.1 | 66.0 |
| InstructCodeT5+ | 16B | 35.0 | - |
| WizardCoder | 15.5B | 57.3 | 51.8 |
| Pangu-Coder 2 | 15.5B | 61.6 | - |
| OctoCoder | 15.5B | 46.2 | - |
| CodeFuse | 34B | 74.4 | - |
| DeepSeek Coder-Instruct | 33B | 79.3 | 70.0 |
| GPT-4 | - | 67.0/82 | - |
| PaLM 2* | S | 37.6 | 50.0 |
| Code LLaMA | 34B | 53.7 | 56.2 |
| Phi-1.5 | 1.3B | 41.4 | 43.5 |

## 5.4 Decoders

After the monumental debut of GPT-3 (Brown et al., 2020) and the discovery of in-context learning, decoder-only Transformer models have become dominant in language modeling (Rae et al., 2021; Hoffmann et al., 2022; Chowdhery et al., 2023; Scao et al., 2022; Touvron et al., 2023a;b, *inter alia*). Many models similarly pretrained with CLM have also emerged in code processing, such as GPT-C (Svyatkovskiy et al., 2020), CodeGPT (Lu et al., 2021), PolyCoder (Xu et al., 2022), CodeGen (Nijkamp et al., 2023b), PyCodeGPT (Zan et al., 2022), Pangu-Coder (Christopoulou et al., 2022), CodeGeeX (Zheng et al., 2023a), Jam (Su et al., 2023a), Phi-1 (Gunasekar et al., 2023), and CodeFuse (Di et al., 2023). Of these models, several alternative training objectives have been experimented with, such as MLM and Masked CLM[7] in Pangu-Coder, but are found to underperform compared with CLM-only training. Zan et al. (2022) also propose continual training on sketches, where the model learns to first generate a sketch of a program and then the actual code. Notably, Gunasekar et al. (2023) present Phi-1, a 1.3B small model trained on a dataset of only 7B tokens consisting of 6B tokens from StackOverflow and 1B synthetic data generated by ChatGPT but achieving 50.6 pass@1 on HumanEval and 55.5 pass@1 on MBPP, comparable to much larger (both in model size and training data size) models such as Code LLaMA or PaLM 2.

---

[7]In their paper, MLM is conducted by replacing tokens in the input with `<mask>` and predicting it from only the left context, while Masked CLM is performed by adding a `<mask>` in the input and predicting the next token from it. Both tasks do not change the attention mask patterns of the model.

Although Christopoulou et al. (2022) report denoising objectives to underperform in decoder-only models, there have been other works that successfully combine denoising or multi-task pretraining with decoder architecture. Incoder (Fried et al., 2023), SantaCoder (Allal et al., 2023), StarCoder (Li et al., 2023i), DeepSeek Coder (Guo et al., 2024), and CodeShell (Xie et al., 2024) are trained with fill-in-the-middle (FIM) objective, also referred to as causal masking by Fried et al. (2023), which is essentially span corruption (Raffel et al., 2020) adopted to decoder-only architecture. One of the visible advantages of these infilling objectives is that they inject the models with the ability to fill in blanks in the middle of input code at inference time, while CLM allows only for autoregressive generation. As Table 4 shows, however, these objectives also lead to higher performance on downstream tasks when compared with CLM-only models such as CodeGen, although the exact benefits of infilling training remain controversial (Nijkamp et al., 2023a).

Observing Table 3, it is clear that decoder-only models for code have generally followed the practices in NLP more closely, when compared with other model architectures. All these models use pre-normalization, while MQA, RoPE, and parallel attention have also been adopted by several models. Notably, the three most recent models - StarCoder, Phi-1, and CodeFuse - also employ FlashAttention to improve model throughput.

### 5.5 UniLMs

Following UniLM (Dong et al., 2019) in NLP, several works in code processing have also pretrained this fourth family of Transformer models on code. CugLM (Liu et al., 2020) is trained with both CLM and MLM + NSP via alternating attention masks, while UniXcoder is trained with CLM, MLM, Span Corruption (in Prefix LM style) along with auxiliary objectives including contrastive learning and text-code mutual generation. Both two models, however, are relatively small in size, and whether or not this architecture is suitable for code processing is yet to be explored at scale.

### 5.6 Diffusion Models

Currently the Transformer architecture dominates text generation, but several works (Li et al., 2022d; Lin et al., 2023) have also adopted Diffusion Models (Ho et al., 2020) from computer vision for text generation. Recently CodeFusion (Singh et al., 2023) also introduces diffusion models into code modeling, and demonstrates that a 75M diffusion model can outperform StarCoder, CodeT5+, and GPT-3 on three code synthesis datasets.

### 5.7 Instruction Finetuning and Reinforcement Learning for Code

In natural language processing, training models on a diverse set of tasks with instruction prefix, known as instruction finetuning, has been shown to unlock the ability of cross-task generalization (Ouyang et al., 2022; Chung et al., 2022; Iyer et al., 2022). At first, these instruction data samples are manually compiled or crowd-sourced (Wei et al., 2022b; Sanh et al., 2022), but later researches find LLM-generated instructions to be sufficient (Wang et al., 2023g; Honovich et al., 2023).

Following these works in natural language, researchers from the code community have applied instruction tuning to their models as well. Wang et al. (2023h) finetune CodeT5+ with 20K instruction data generated by InstructGPT (Ouyang et al., 2022) to obtain InstructCodeT5+. WizardCoder (Luo et al., 2023) follows the methods of WizardLM (Xu et al., 2024) to evolve 20K code Alpaca (Taori et al., 2023) samples into a 78K dataset and uses it to finetune StarCoder. Pangu-Coder 2 (Shen et al., 2023) also uses WizardLM's Evol-Instruct to generate 68K instruction samples from 20K code Alpaca, but also introduces reinforcement learning via Rank Responses to align Test & Teacher Feedback (RRTF). OctoCoder (Muennighoff et al., 2024), on the other hand, takes a different path and uses Git commit histories as instruction data to finetune StarCoder and CodeGeeX2. More recently, CodeFuse (Liu et al., 2023b) also employs multitask-finetuning and explicitly introduces multiple downstream tasks into their instruction data. The performance of these instruction finetuned code models can also be found in Table 4.

In NLP, another technology closely related to instruction finetuning is reinforcement learning from human feedback (RLHF), which has played a significant role in aligning LLMs with human values (Ouyang et al., 2022; Bai et al., 2022). The merit of reinforcement learning is that it can incorporate non-differentiable

reward signals into training, such as BLEU (Bahdanau et al., 2017) and human preference (Christiano et al., 2017), but the human feedback required in aligning LLMs often involves extensive labor on annotation. In comparison, applying reinforcement learning to code models has a natural advantage, as compilers can be used for automatically generating feedback for code samples produced by language models.

CodeRL (Le et al., 2022) is one such model, which defines four levels of rewards for each generated program (viz. compile error, runtime error, unit test failure, pass) as well as fine-grained token-level reward estimated by a critic model. The actor model, which is an extension of CodeT5, is then trained with REINFORCE algorithm (Williams, 1992). Similarly, CompCoder (Wang et al., 2022d) and PPOCoder (Shojaee et al., 2023) train CodeGPT and CodeT5 respectively with proximal policy optimization (Schulman et al., 2017), while RLTF (Liu et al., 2023d) proposes fine-grained feedback based on the error information and location provided by the compiler, as well as adaptive feedback that takes the ratio of passed test cases into account.

# 6 Code Features for Language Models

A major difference between programming languages and natural languages is that the former is artificially defined to be precise and unambiguous, and need to be compiled (or interpreted) without error before execution. This allows for a much larger flexibility in designing pretraining objectives on code, beside lexical manipulations such as CLM, MLM, and Span Corruption. A similar trend can be observed in the last years before neural networks were introduced into mainstream NLP literature (Sutskever et al., 2014; Bahdanau et al., 2015), when researchers in the MT community utilized alternative views of text such as syntactic features to improve the performance of SMT systems (Galley et al., 2006; Chiang, 2007). These features, however, are not universally applicable or even agreed upon, and often result in highly complicated systems (for example, the size of English part-of-speech tagging's label set may range from dozens to hundreds).

Programming languages, however, fare much better in these aspects. Each mainstream programming language, such as C, Python, and Java, comes with readily available compiler toolkits that allow for easy and accurate extraction of semantic information such as Abstract Syntax Tree (AST), language-independent Intermediate Representation (IR), and auxiliary information such as type of each token and control/data flow graph (CFG/DFG). Thus, in the context of Transformer-based language modeling for code, many works have incorporated these features into their training procedure.

## 6.1 Abstract Syntax Tree and Intermediate Representation

AST is one of the most common intermediate results of the compiling process, where a program is parsed into a tree of operations and their operands. Before the popularization of Transformer in the code processing community, there had been works such as InferCode (Bui et al., 2021a) that processes these representations with special network architectures like Tree-Based CNN and conducts self-supervised pretraining by predicting subtrees.

TreeBERT (Jiang et al., 2021b) is one of the first attempts to take AST into the Transformer-based pretraining-finetuning framework. It's a Transformer encoder-decoder pretrained with Tree MLM and Node Order Prediction, where the encoder takes a set of constituent paths in the AST as input (with each token being a path, which is the concatenation of its nodes' representations) while the decoder takes the code as input. Tree MLM is then performed by masking certain nodes in a path representation and its corresponding code tokens in the decoder input, while Node Order Prediction is accomplished by swapping nodes in a path and predicting it with a `[CLS]` token similar to BERT.

The method used by TreeBERT, however, is complicated and does not scale well. Later works mostly opt to first process AST into a text sequence and treat it like a normal part of the input. Wang et al. (2021d), for example, process AST with depth-first traversal and concatenate it with code and comment, and then train SynCoBERT (which, unlike TreeBERT, is actually a BERT-like encoder-only model) with four objectives: 1) MLM; 2) identifier tagging; 3) AST edge prediction (predicting whether there exists an edge between two AST nodes from the dot product of these nodes' representations); and 4) contrastive learning over i) code and AST pairs, as well as ii) text and code-AST pairs. Similarly, SPT-Code (Niu et al., 2022), a Transformer encoder-decoder, takes the concatenation of code, sequentialized AST, and text as input, and

is pretrained with 1) span corruption; 2) code-AST prediction (NSP with one segment being code and one segment being AST); and 3) method name generation, a special form of span corruption where a method name is masked. Different from other works, however, they do not take the docstrings as the text segment in their input, but instead concatenate all method names appearing in the code as a succinct natural language description. Likewise, UniXcoder (Guo et al., 2022) takes flattened AST instead of source code as its input during training.

In the compiling pipeline, AST is usually followed by language-independent intermediate representations, such as LLVM IR (Lattner & Adve, 2004). Such features' independence from specific programming languages makes them suitable candidates for translation pivots, as is English in machine translation of low-resource natural languages (Leng et al., 2019). Szafraniec et al. (2023) take advantage of this characteristic and extend Transcoder (Rozière et al., 2020) with translation language modeling (Conneau & Lample, 2019) over code and IR, as well as IR generation from code. They also investigate other objectives such as IR decompilation (i.e. generating code from IR) and IR pivot (i.e. directly generating code in one language from the IR of another language), both showing promising results.

## 6.2   Control Flow and Data Flow

While AST and IR have proved to be useful information in certain tasks such as code translation, they are static by nature, just like the source code, and may fail to capture semantic properties of code that are only revealed at runtime (Wang & Su, 2020). Such semantics, however, are contained in dynamic features such as control flow and data flow. Similar to AST, specialized networks were used to process such information before the rise of pretrained Transformers, such as Message Passing Neural Network used by ProGraML (Cummins et al., 2021). Unlike AST, however, even after pretrained Transformers became dominant few works have looked in this direction.

GraphCodeBERT (Guo et al., 2021a) is one of such works, which creates special tokens and position embeddings for variables in the flow graph, and concatenates the variable sequence after text and source code to construct model input, with tailored attention masks on the code and variable segments: tokens from code segment and variable segment can attend to each other if and only if the variable is identified from the code token, and for tokens within the variable segment, $v_i$ is allowed to attend to $v_j$ if there is a direct edge from $v_j$ to $v_i$ in the dataflow. The model is then pretrained with MLM in combination with edge prediction and node alignment, both of which are accomplished by binary classification from the dot product of two tokens' representations (one from code segment and one from variable segment for node alignment, and both from variable segment for edge prediction).

## 6.3   Type

Apart from AST, IR, and data flow, type information has also been used to aid language models in processing code. CugLM (Liu et al., 2020), for example, uses type information during finetuning to aid in the prediction of tokens for unidirectional MLM (i.e. MLM with unidirectional attention mask): the type of a masked token is first predicted from the final Transformer layer's representation, and then the token itself is predicted based on both the hidden representation and predicted type. In contrast, both CodeT5 (Wang et al., 2021f) and SynCoBERT (Wang et al., 2021d) include identifier tagging in their pretraining objectives, which can be viewed as coarse-grained type prediction.

## 6.4   Program Transformation

As we have shown in Section 5.3, function-preserving program transformations have proven to be important techniques in pretraining code language models. Obfuscation is one instance of program transformation, and others include loop transformation (for-while), condition transformation (if-switch), dead code injection (e.g. `if True:  pass`), and statement swapping. DOBF (Lachaux et al., 2021) and NatGen (Chakraborty et al., 2022a) are two code language models pretrained to recover the original program from such transformations, while Wang et al. (2022a) also apply program transformations during the finetuning stage of language models to make them more robust to transformed test samples.

Notably, Wang et al. (2022e) integrate many of the aforementioned features into Code-MVP: source code, docstrings, AST, CFG, and transformed source code via identifier renaming, loop exchange, and dead code insertion. The model, initialized from GraphCodeBERT, is then trained with MLM, fine-grained type prediction, and contrastive learning across different views, such as text vs. code, code vs. AST, and code vs. CFG.

# 7 LLMs in Software Development

As language models set new records on software engineering benchmarks, software engineering technologies are also expanding the boundaries of language models in return, and have subsequently led them into real-world development cycles.

## 7.1 LLMs Extended with Coding Tools

Research in the NLP community has shown that LLMs can learn to use external tools such as calculators, MT systems, and search engines (Thoppilan et al., 2022; Schick et al., 2023). As such, *interpreter* has been used to augment LLMs in complex reasoning tasks. PAL (Gao et al., 2023b) and PoT (Chen et al., 2023e) both extend Codex with Python interpreters for numerical calculations, while ViperGPT (Surís et al., 2023) extends it further by calling vision APIs to extract information from visual input and answer related questions.

However, LLMs do not always produce code that can be interpreted and executed, and there has also been a line of works that explore the possibility of emulating the interpreter with LLMs themselves. Nye et al. (2021) trains LLMs to emulate the execution of programs by outputting program states at each step. More recently, Li et al. (2023a) propose Chain-of-Code, where a real interpreter executes the generated code until an error occurs, whereupon the LLM takes over to simulate execution. Chae et al. (2024) similarly propose Think-and-Execute framework, where an LLM generates pseudo code and then simulates its execution to solve reasoning tasks.

Apart from alleviating the burden of numerical calculation in abstract reasoning tasks, interpreter (together with unit tests) also provides feedback on the process of code generation itself, allowing for interactive generation and refinement of code. Such works include Self-Edit (Zhang et al., 2023c), LeTI (Wang et al., 2023f), OpenCodeInterpreter (Zheng et al., 2024), ProCoder (Bi et al., 2024), Cycle (Ding et al., 2024), and SOAP (Huang et al., 2024), which run model-generated code against unit tests to provide feedback for further refinement. Alternatively, CodeT (Chen et al., 2023a), TiCoder (Bareiß et al., 2022), and CONLINE (He et al., 2024) also utilize the LLM itself to generate unit tests, while Self-Refine (Madaan et al., 2023) sends the generated code to an LLM instead of an interpreter for feedback. Zhou et al. (2023a) show that OpenAI's interpreter plugin[8] allows GPT-4 to self-debug, while InterCode (Yang et al., 2023d) provides a benchmark for evaluating interactive coding. In Section 5.7 we have also shown that the execution results on unit tests serve as natural supervision signals for reinforcement learning on code.

A topic closely related to tool using in LLM research is *planning* as intelligent agents, which has been shown to enhance LLMs' capability both theoretically and empirically (Feng et al., 2023a). Ruan et al. (2023) find that LLMs can plan to solve complex tasks using external SQL generators and Python generators, while CodePlan (Bairi et al., 2023) demonstrates they can perform repository-level coding via adaptive planning.

Another stream of works use LLMs to create multi-agent systems for code generation, such as self-collaboration (Dong et al., 2023), ChatDev (Qian et al., 2023), MetaGPT (Hong et al., 2023), LCG (Lin et al., 2024), MAGIS (Tao et al., 2024), and SoA (Ishibashi & Nishimura, 2024). In these frameworks, multiple LLMs are prompted to play distinct roles such as programmer, reviewer, and manager. These roles interact with each other, breakdown code generation into different phases (e.g. designing, coding, testing, and documenting), and collaborate to complete complex tasks.

---

[8]https://openai.com/blog/chatgpt-plugins#code-interpreter

## 7.2 LLMs Integrated into Software Development

With the increase in LLMs' interactive coding capability, researchers have also started to integrate them into each and every process of software development.

Auto code completion is one of the earliest applications of language models in software development, as they require only the ability to predict the next token. Even before language models scaled to billions of parameters, there had been integration of completion systems such as Pythia (Svyatkovskiy et al., 2019) and IntelliCode (Svyatkovskiy et al., 2020) into popular IDEs.

Recently, however, the application of code language models have transcended simple code completion. GitHub Copilot is arguably one of the most popular AI code assistants, with diverse features including code generation, vulnerability detection, and license management[9], while CodeFuse (Di et al., 2023) also integrates code generation, code translation, code commenting, and testcase generation into a single IDE extension. As code language models become larger, however, their client-side deployment and real-time performance also raise new challenges.

As LLMs continue to advance, building applications on top of them is also evolving into a consequential task itself. Many open-source frameworks for such applications have been released, including LangChain[10], AutoGPT[11], and WorkGPT[12]. These frameworks provide abstractions over language models for developers, and are actively revolutionizing the entire process of software development even as this survey is being finalized.

## 7.3 Analysis of LLM-Generated Code

As AI code assistants become prevalent, many recent works have also focused on examining AI-generated code from different aspects, including correctness (Nguyen & Nadi, 2022; Yetistiren et al., 2023), bugs (Jesse et al., 2023; Tambon et al., 2024), vulnerabilities (Asare et al., 2023; Sandoval et al., 2023; Perry et al., 2023; Hamer et al., 2024), syntactic robustness (Sarker et al., 2024), efficiency (Niu et al., 2024), and hallucinations (Liu et al., 2024). Asare et al. (2023) and Sandoval et al. (2023)'s studies show that AI code assistants do not introduce extra security risks compared with human programmers, and Hamer et al. (2024) also find that ChatGPT's responses to security-related questions contain less vulnerabilities than answers from StackOverflow. Contrarily, Perry et al. (2023) find that users write significantly less secure code when assisted by AI. In terms of code complexity, Nguyen & Nadi (2022) find GitHub Copilot to generate low-complexity code with no statistically significant differences between languages, whereas Liu et al. (2023n) find ChatGPT to generate the most complex code in C and the least complex code in Python. Overall, AI code assistants are still in their nascent stage and constantly evolving, and their implications on software development are yet to be investigated systematically.

## 8 Conclusion and Challenges

In this work, we systematically reviewed the history of pretrained Transformer language models in code processing and other software engineering tasks. The advancement in code modeling generally follows the history course of NLP, evolving from SMT models, to NMT models, and then to finetuning pretrained Transformers and lastly to few-shot application of LLMs and even autonomous agents in real-world production. Unlike natural languages, the nature of code makes it easy to extract auxiliary information from alternative views, and to utilize interpreter and unit tests for automatic feedback.

With these in mind, we identify several challenges in the current development of code modeling.

- **More comprehensive and challenging benchmarks**. The widely used HumanEval benchmark plays a key role in the evolution of Code LLMs. However, it is relatively small and its scoreboard has been manipulated to near perfect, and the community is eager for a new standard to evaluate LLMs. HumanEval and

---

[9]https://github.com/features/copilot
[10]https://www.langchain.com/
[11]https://github.com/Significant-Gravitas/AutoGPT
[12]https://github.com/team-openpm/workgpt

other similar benchmarks focus on generating standalone Python functions from well-structured docstrings, which do not reflect real-world user behaviors. In real-world scenarios, software requirements are seldom condensed into a single docstring, and LLMs must learn to communicate effectively for clarifications when uncertain about the requirements (Wu & Fard, 2024). Also, standalone functions are hardly used in production, where software has complex dependencies within a repository. Thus, next-generation benchmarks should also take such cross-file context into account, and recent benchmarks such as SWE-bench (Jimenez et al., 2023) represent promising efforts in this direction. Contrary to the models' perfect scores on HumanEval, these repository-level benchmarks expose LLMs' limits in real-world applications, with the most recent SWE-agent resolving only 12.5% real-world GitHub issues in SWE-bench (Yang et al., 2024b).

- **Evaluation and application of LLMs beyond traditional code generation**. As we have pointed out in Section 2, current evaluation of LLMs' coding capability in the NLP community is focused on code generation, while overlooking other activities in software engineering, such as software modeling and testing. From an application point-of-view, the most widespread application of LLMs in software engineering is IDE plugins, which provide developers with code suggestions, while other stages in software development that we mentioned in Section 2 are also largely overlooked. However, it should be also noted that LLMs trained on code are highly specialized to reasoning-heavy tasks such as programming and math, and non-coding tasks in SE - such as requirement elicitation, product management, and marketing - are probably better suited for general-domain models. Beyond the previously mentioned text-based evaluation and applications, the recent advancement of multimodel LLMs also created new opportunities, especially for UI design and other activities that involve visual input, leading to novel tasks such as visually grounded code generation (Li et al., 2024b; Wu et al., 2024a), webpage reverse engineering (Si et al., 2024; Laurençon et al., 2024), and layout design (Feng et al., 2023b).

- **Acquisition of high-quality data**. With Gunasekar et al. (2023) achieving SOTA performance with a 1.3B model trained on textbook data, we believe the selection of training data and utilization of synthetic data will be ever more prominent in the near future, for both self-supervised pretraining and supervised finetuning.

- **Integration of code features into language models**. As we noted in Section 6.2, CFG and DFG are yet to be employed at scale in code language modeling. The few works that do employ data flow make changes to the models' attention masks, which severely limits their cross-task generalization and scaling ability. We believe the seamless integration of such features into textual input is worth researching in the future.

- **Beyond the imperative programming paradigm**. In Section 2.1.7 we have shown that most of the current research on code LLM focus on popular imperative programming languages such as C and Python, while neglecting declarative and functional languages except SQL. However, the paradigm of declarative and functional languages makes them more aligned to natural languages, and thus is worth more research attention in the future.

- **Alternative model architectures and training objectives**. In Table 3, we have shown that many code language models are pretrained with auxiliary objectives specific to code, but these models all belong to the encoder-only or encoder-decoder family, while decoder-only models are yet to be augmented with alternative objectives. Also, as pioneered by Singh et al. (2023), we believe diffusion models will find its ground in code modeling in the future.

- **Safety and ethics issues related to code LLMs**. As language models grow in might, they also raise safety concerns including but not limited to data contamination, toxic or biased generation, personal information leak, and hallucinations. In software development, these models should be deployed with extra caution, as their generated code may contain security risks leading to catastrophic results. Pretraining data is also becoming a sensitive topic of ethics, and Kocetkov et al. (2023) take a meaningful step towards this issue by allowing developers to remove their code from the Stack. As synthetic training data becomes widespread, researchers should also proceed with caution about such practice, as the consequence of training AI models with AI generated data is yet to be investigated at scale.

With the presentation of this survey, we hope to provide a global view of language models' application in software engineering and connect the research from the two communities. We believe the current surge of LLMs will be ultimately transformed into real-world applications, and lead humanity into a brighter future.

**Acknowledgments**

This work is supported by Ant Group. Ziyin Zhang and Rui Wang are partially supported by the National Natural Science Foundation of China (62176153) and the Shanghai Municipal Science and Technology Major Project (2021SHZDZX0102, as the MoE Key Lab of Artificial Intelligence, AI Institute, Shanghai Jiao Tong University).

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

# A  Benchmarks for Downstrem Tasks

Table 5, 6, 7, 8, 9, 10 list benchmark datasets for code downstream tasks. For the size of these datasets, we use self-reported number whenever available.

Table 5: Benchmarks for text-to-SQL generation.

| Task | Date | Benchmark | Source | Size |
|---|---|---|---|---|
| Text-to-SQL | 1990 | ATIS | Hemphill et al. (1990); Dahl et al. (1994) | 11508 |
| | 1996 | GeoQuery | Zelle & Mooney (1996) | 877 |
| | 2000 | Restaurants | Tang & Mooney (2000) | 378 |
| | 2014-09 | MAS | Li & Jagadish (2014) | 196 |
| | 2017-02 | Yelp | Yaghmazadeh et al. (2017) | 128 |
| | 2017-02 | IMDb | Yaghmazadeh et al. (2017) | 131 |
| | 2017-04 | Scholar | Iyer et al. (2017) | 816 |
| | 2017-08 | WikiSQL | Zhong et al. (2017) | 80654 |
| | 2018-06 | Advising | Finegan-Dollak et al. (2018) | 4570 |
| | 2018-09 | Spider | Yu et al. (2018c) | 10181 |
| | 2019-06 | SParC | Yu et al. (2019c) | 12726 |
| | 2019-07 | MIMICSQL | Wang et al. (2020b) | 10000 |
| | 2019-09 | CoSQL | Yu et al. (2019b) | 15598 |
| | 2020-05 | Criteria-to-SQL | Yu et al. (2020) | 2003 |
| | 2020-10 | Squall | Shi et al. (2020) | 11276 |
| | 2020-10 | Spider-Realistic | Deng et al. (2021) | 508 |
| | 2021-06 | Spider-Syn | Gan et al. (2021a) | 8034 |
| | 2021-06 | SEDE | Hazoom et al. (2021) | 12023 |
| | 2021-06 | KaggleDBQA | Lee et al. (2021) | 400 |
| | 2021-09 | Spider-DK | Gan et al. (2021b) | 535 |
| | 2022-05 | Spider-SS | Gan et al. (2022) | 8034 |
| | 2022-05 | Spider-CG | Gan et al. (2022) | 45599 |
| | 2023-05 | BIRD | Li et al. (2023g) | 12751 |

Table 6: Benchmarks for program synthesis. JS is short for JavaScript. *Automatically mined/human-annotated. †These are 1749 prompts for 48 problems. ‡10538 prompts for 1420 methods. ◇Machine-generated/human-annotated prompts.

| Task | Date | Benchmark | Source | Size | Language |
|---|---|---|---|---|---|
| | 2018-02 | NL2Bash | Lin et al. (2018b) | 9305 | Bash |
| | 2018-08 | CONCODE | Iyer et al. (2018) | 104K | Java |
| | 2019-10 | JuICe | Agashe et al. (2019) | *1.5M/3725 | Python |
| | 2021-05 | APPS | Hendrycks et al. (2021a) | 10000 | Python |
| | 2021-07 | HumanEval | Chen et al. (2021b) | 164 | Python |
| | 2021-08 | MBPP | Austin et al. (2021) | 974 | Python |
| | 2021-08 | MathQA-Python | Austin et al. (2021) | 23914 | Python |
| | 2021-08 | PlotCoder | Chen et al. (2021c) | 40797 | Python |
| | 2022-01 | DSP | Chandel et al. (2022) | 1119 | Python |
| | 2022-02 | CodeContests | Li et al. (2022h) | 13610 | C++, Python, Java |
| | 2022-03 | MCoNaLa | Wang et al. (2023i) | 896 | Python |
| Program Synthesis | 2022-06 | AixBench | Hao et al. (2022) | 336 | Java |
| | 2022-10 | MBXP | Athiwaratkun et al. (2023) | 12.4K | 13 |
| | 2022-10 | Multilingual HumanEval | Athiwaratkun et al. (2023) | 1.9K | 12 |
| | 2022-10 | MathQA-X | Athiwaratkun et al. (2023) | 5.6K | Python, Java, JS |
| | 2022-11 | ExeDS | Huang et al. (2022b) | 534 | Python |
| | 2022-11 | DS-1000 | Lai et al. (2023) | 1000 | Python |
| | 2022-12 | ODEX | Wang et al. (2022h) | 945 | Python |
| | 2023-02 | CoderEval | Yu et al. (2023a) | 460 | Python, Java |
| | 2023-03 | xCodeEval | Khan et al. (2023) | 5.5M | 11 |
| | 2023-03 | HumanEval-X | Zheng et al. (2023a) | 820 | Python, C++, Java, JS, Go |
| | 2023-06 | StudentEval | Babe et al. (2023) | †1749 | Python |
| | 2023-06 | DotPrompts | Agrawal et al. (2023) | ‡10538 | Java |
| | 2023-09 | CodeApex | Fu et al. (2023) | 476 | C++ |
| | 2023-09 | VerilogEval | Liu et al. (2023g) | ◇8645/156 | Verilog |
| | 2023-11 | ML-Bench | Liu et al. (2023l) | 10040 | Bash |
| | 2024-01 | ParEval | Nichols et al. (2024) | 420 | C++, HIP, CUDA |
| | 2024-04 | MMCode | Li et al. (2024b) | 3548 | Python |
| | 2024-04 | PECC | Haller et al. (2024) | 2396 | Python |
| | 2024-05 | NaturalCodeBench | Zhang et al. (2024b) | 402 | Python, Java |
| | 2024-05 | Plot2Code | Wu et al. (2024a) | 132 | Python |
| | 2024-05 | MHPP | Dai et al. (2024b) | 140 | Python |

Table 7: Benchmarks for code translation and program repair. JS is short for JavaScript. *These are pairwise sample counts. For example, HumanEval-X includes 164 programs, each implemented in 5 languages, totaling $164 \times (5 \times 4 / 2) = 1640$ translation pairs. †These are code change datasets, and only a subset therein concern bug-fixing.

| Task | Date | Benchmark | Source | Size | Language |
|------|------|-----------|--------|------|----------|
| Code Translation | 2020-06 | GeeksforGeeks | Rozière et al. (2020) | 1.4K | C++, Java, Python |
| | 2021-02 | CodeTrans | Lu et al. (2021) | 11.8K | Java, C# |
| | 2021-08 | Avatar | Ahmad et al. (2023) | 9515 | Java, Python |
| | 2022-06 | CoST | Zhu et al. (2022b) | *132K | C++, Java, Python, C#, JS, PHP, C |
| | 2022-06 | XLCoST | Zhu et al. (2022a) | *567K | C++, Java, Python, C#, JS, PHP, C |
| | 2023-03 | xCodeEval | Khan et al. (2023) | *5.6M | 11 |
| | 2023-03 | HumanEval-X | Zheng et al. (2023a) | *1640 | Python, C++, Java, JS, Go |
| | 2023-08 | G-TransEval | Jiao et al. (2023) | *4000 | C++, Java, C#, JS, Python |
| | 2023-10 | CodeTransOcean | Yan et al. (2023) | 270.5K | 45 |
| Program Repair | 2014-07 | Defects4J | Just et al. (2014) | 357 | Java |
| | 2015-12 | ManyBugs | Goues et al. (2015) | 185 | C |
| | 2015-12 | IntroClass | Goues et al. (2015) | 998 | C |
| | 2016-11 | BugAID | Hanam et al. (2016) | 105K | JS |
| | 2017-02 | DeepFix | Gupta et al. (2017) | 6971 | C |
| | 2017-05 | Codeflaws | Tan et al. (2017) | 3902 | C |
| | 2017-10 | QuixBugs | Lin et al. (2017) | 80 | Java, Python |
| | 2018-05 | Bugs.jar | Saha et al. (2018) | 1158 | Java |
| | 2018-12 | BFP | Chakraborty et al. (2022b) | 124K | Java |
| | 2019-01 | Bears | Madeiral et al. (2019) | 251 | Java |
| | 2019-01 | unnamed | Tufano et al. (2019a) | †21.8K | Java |
| | 2019-04 | BugsJS | Gyimesi et al. (2019) | 453 | JS |
| | 2019-05 | BugSwarm | Tomassi et al. (2019) | 1827/1264 | Java/Python |
| | 2019-05 | CPatMiner | Nguyen et al. (2019) | †17K | Java |
| | 2019-05 | ManySStuBs4J | Karampatsis & Sutton (2020) | 154K | Java |
| | 2019-11 | Refactory | Hu et al. (2019) | 1783 | Python |
| | 2020-07 | CoCoNut | Lutellier et al. (2020) | 24M | Java, JS, C, Python |
| | 2020-10 | Review4Repair | Huq et al. (2022) | 58021 | Java |
| | 2020-11 | BugsInPy | Widyasari et al. (2020) | 493 | Python |
| | 2021-07 | TFix | Berabi et al. (2021) | 105K | JS |
| | 2021-08 | Megadiff | Monperrus et al. (2021) | †663K | Java |
| | 2022-01 | SSB/TSSB | Richter & Wehrheim (2022) | 9M/3M | Python |
| | 2022-10 | FixJS | Csuvik & Vidács (2022) | 324K | JS |
| | 2022-11 | TypeBugs | Oh & Oh (2022) | 93 | Python |
| | 2023-03 | xCodeEval | Khan et al. (2023) | 4.7M | 11 |
| | 2023-04 | RunBugRun | Prenner & Robbes (2023) | 450K | C, C++, Java, Python, JS, Ruby, Go, PHP |
| | 2023-08 | HumanEvalPack | Muennighoff et al. (2024) | 984 | Python, JS, Go, Java, C++, Rust |
| | 2024-01 | DebugBench | Tian et al. (2024) | 4253 | C++, Java, Python |

Table 8: Benchmarks for code summarization, code completion, commit message generation, and defect/vulnerability detection. JS is short for JavaScript. *The task of code completion can be evaluated on any source code corpus, so we only list a few widely used benchmarks here. For cross-file code completion please refer to Table 10. †With/without verb-direct object filter.

| Task | Date | Benchmark | Source | Size | Language |
|------|------|-----------|--------|------|----------|
| Code Summarization | 2016-08 | CODE-NN | Iyer et al. (2016) | 66K/32K | C#/SQL |
| | 2017-07 | unnamed | Barone & Sennrich (2017) | 150K | Python |
| | 2018-05 | DeepCom | Hu et al. (2018a) | 588K | Java |
| | 2018-07 | TL-CodeSum | Hu et al. (2018b) | 411K | Java |
| | 2018-11 | unnamed | Wan et al. (2018) | 109K | Python |
| | 2019-02 | unnamed | LeClair et al. (2019) | 2.1M | Java |
| | 2019-09 | CodeSearchNet | Husain et al. (2019) | 2.3M | Go, JS, Python, PHP, Java, Ruby |
| | 2023-08 | HumanEvalPack | Muennighoff et al. (2024) | 984 | Python, JS, Go, Java, C++, Rust |
| *Code Completion | 2013-05 | GitHub Java Corpus | Allamanis & Sutton (2013) | 2.1M | Java |
| | 2016-10 | Py150 | Raychev et al. (2016a) | 150K | Python |
| | 2016-10 | JS150 | Raychev et al. (2016a) | 150K | JS |
| | 2023-06 | LCC | Guo et al. (2023b) | 360K | Python, Java, C# |
| Commit Message Generation | 2017-03 | unnamed | Jiang & McMillan (2017) | 509K | Java |
| | 2017-04 | CommitGen | Loyola et al. (2017) | 153K | Python, C++, Java, JS |
| | 2017-08 | CommitGen | Jiang et al. (2017) | †32K/75K | Java |
| | 2018-09 | NNGen | Liu et al. (2018) | 27K | Java |
| | 2019-05 | PtrGNCMsg | Liu et al. (2019c) | 64.9K | Java |
| | 2019-08 | CoDiSum | Xu et al. (2019a) | 90.7K | Java |
| | 2019-12 | ATOM | Liu et al. (2022c) | 160K | Java |
| | 2021-05 | CommitBERT | Jung (2021) | 346K | Python, PHP, Go, Java, JS, Ruby |
| | 2021-07 | MCMD | Tao et al. (2021) | 2.25M | Java, C#, C++, Python, JS |
| | 2021-07 | CoRec | Wang et al. (2021c) | 107K | Java |
| | 2023-07 | ExGroFi | Wang et al. (2023c) | 19263 | Java |
| | 2023-08 | CommitChronicle | Eliseeva et al. (2023) | 10.7M | 20 |
| Defect (Vulnerability) Detection | 2018-01 | CGD | Li et al. (2018b) | 62K | C, C++ |
| | 2018-07 | Draper VDISC | Russell et al. (2018) | 12.8M | C, C++ |
| | 2018-07 | SySeVR | Li et al. (2022j) | 15591 | C, C++ |
| | 2018-04 | unnamed | Lin et al. (2018a) | 32988 | C, C++ |
| | 2019-02 | unnamed | Ponta et al. (2019) | 624 | Java |
| | 2019-09 | Devign | Zhou et al. (2019) | 48687 | C |
| | 2019-11 | unnamed | Lin et al. (2021) | 170K | C, C++ |
| | 2019-12 | GREAT | Hellendoorn et al. (2020) | 2.8M | Python |
| | 2020-01 | MVD | Zou et al. (2021) | 182K | C, C++ |
| | 2020-02 | unnamed | Lin et al. (2019) | 1471 | C |
| | 2020-09 | ReVeal | Chakraborty et al. (2022c) | 18K | C |
| | 2020-09 | Big-Vul | Fan et al. (2020) | 265K | C, C++ |
| | 2021-02 | D2A | Zheng et al. (2021) | 1.3M | C, C++ |
| | 2021-05 | PyPIBugs | Allamanis et al. (2021) | 2374 | Python |
| | 2021-07 | CVEfixes | Bhandari et al. (2021) | 5495 | 27 |
| | 2021-08 | CrossVul | Nikitopoulos et al. (2021) | 27476 | 40+ |
| | 2023-04 | DiverseVul | Chen et al. (2023g) | 349K | C, C++ |
| | 2023-06 | VulnPatchPairs | Risse & Böhme (2023) | 26K | C |
| | 2023-11 | VulBench | Gao et al. (2023d) | 455 | C |

Table 9: Benchmarks for code retrieval, code reasoning, type inference, and clone detection/code search,. JS is short for JavaScript. *These benchmarks include a large number of automatically constructed samples, and a small set of human-annotated samples. †These are general-domain reasoning benchmarks, and only a subset therein concern programming, algorithms, and other topics related to computer science. ‡These are project counts (or, in the case of Cassano et al. (2023c), file counts). Yee & Guha (2023) propose to measure project-level type check rate instead of type prediction accuracy for TypeScript. ⋄These are 21K/24K Java/Python methods for 576 programming problems.

| Task | Date | Benchmark | Source | Size | Language |
|---|---|---|---|---|---|
| Code Retrieval | 2018-03 | StaQC | Yao et al. (2018) | 268K | Python, SQL |
| | 2018-05 | DeepCS | Gu et al. (2018) | 16M | Java |
| | 2018-05 | CoNaLa | Yin et al. (2018) | *600K/2.9K | Python |
| | 2019-08 | unnamed | Li et al. (2019b) | 287 | Java |
| | 2019-09 | CodeSearchNet | Husain et al. (2019) | *2.3M/99 | Go, JS, Python, PHP, Java, Ruby |
| | 2020-02 | CosBench | Yan et al. (2020) | 52 | Java |
| | 2020-08 | SO-DS | Heyman & Cutsem (2020) | 2.2K | Python |
| | 2020-10 | FB-Java | Ling et al. (2021) | 249K | Java |
| | 2021-02 | AdvTest | Lu et al. (2021) | 280K | Python |
| | 2021-02 | WebQueryTest | Lu et al. (2021) | 1K | Python |
| | 2021-05 | CoSQA | Huang et al. (2021) | 21K | Python |
| | 2024-03 | ProCQA | Li et al. (2024c) | 5.2M | C, C++, Java, Python, Ruby, Lisp, JS, C#, Go, Rust, PHP |
| Code Reasoning | 2020-09 | MMLU | Hendrycks et al. (2021b) | †15908 | |
| | 2021-09 | CodeQA | Liu & Wan (2021) | 120K/70K | Java/Python |
| | 2022-10 | CS1QA | Lee et al. (2022) | 9237 | |
| | 2023-05 | C-Eval | Huang et al. (2023b) | †13948 | |
| | 2023-06 | CMMLU | Li et al. (2023b) | †11528 | |
| | 2023-09 | CodeApex | Fu et al. (2023) | 250 | C++ |
| | 2024-01 | CRUXEval | Gu et al. (2024) | 800 | Python |
| | 2024-05 | PythonIO | Zhang et al. (2024d) | 2650 | Python |
| Type Inference | 2019-12 | TypeWriter OSS | Pradel et al. (2020) | 208K | Python |
| | 2020-04 | Typilus | Allamanis et al. (2020) | 252K | Python |
| | 2020-04 | LambdaNet | Wei et al. (2020) | ‡300 | TypeScript |
| | 2021-04 | ManyTypes4Py | Mir et al. (2021) | 869K | Python |
| | 2022-10 | ManyTypes4TypeScript | Jesse & Devanbu (2022) | 9.1M | TypeScript |
| | 2023-02 | TypeWeaver | Yee & Guha (2023) | ‡513 | TypeScript |
| | 2023-03 | BetterTypes4Py | Wei et al. (2023) | 608K | Python |
| | 2023-03 | InferTypes4Py | Wei et al. (2023) | 4.6K | Python |
| | 2023-05 | OpenTau | Cassano et al. (2023c) | ‡744 | TypeScript |
| Clone Detection / Code Search | 2014-09 | BigCloneBench | Svajlenko et al. (2014) | 6M | Java |
| | 2014-09 | POJ-104 | Mou et al. (2016) | 52K | C, C++ |
| | 2019-05 | unnamed | Perez & Chiba (2019) | ⋄21K/24K | Java/Python |
| | 2019-11 | CLCDSA | Nafi et al. (2019) | 78K | Java, C#, Python |

Table 10: Benchmarks for unit test/assertion generation, log parsing, and repository level coding. *LogHub (2023) is an annotated subset of LogHub (2018). [†]Line Completion/API Invocation Completion/Function Completion. [‡]Retrieval/Completion/Pipeline. *File count. $^\diamond$Migration/Temporal Edit.

| Task | Date | Benchmark | Source | Size | Language |
|---|---|---|---|---|---|
| Unit Test / | 2014-12 | SF110 | Fraser & Arcuri (2014) | 24K | Java |
| Assertion | 2020-09 | Method2Test | Tufano et al. (2021a) | 781K | Java |
| Generation | 2021-08 | ConTest | Villmow et al. (2021) | 365K | Java |
| Log Parsing | 2018-11 | LogHub (2018) | Zhu et al. (2019); He et al. (2020) | 379M | |
| | 2023-08 | LogHub (2023) | Jiang et al. (2023e) | *50.4M | |
| Repository-Level Coding | 2023-03 | RepoEval | Zhang et al. (2023b) | [†]1600/1600/373 | Python |
| | 2023-06 | RepoBench | Liu et al. (2023j) | [‡]890K/9M/43K | Python, Java |
| | 2023-06 | PragmaticCode | Agrawal et al. (2023) | *880 | Java |
| | 2023-06 | Stack-Repo | Shrivastava et al. (2023a) | 816K | Java |
| | 2023-09 | CodePlan | Bairi et al. (2023) | $^\diamond$645/21 | $^\diamond$C#/Python |
| | 2023-10 | SWE-Bench | Jimenez et al. (2023) | 2294 | Python |
| | 2023-10 | CrossCodeEval | Ding et al. (2023) | 9928 | Python, Java, TypeScript, C# |
| | 2024-03 | EvoCodeBench | Li et al. (2024a) | 275 | Python |

# B   Results on Downstream Benchmarks

For the benchmarks listed in Appendix A, we also list code language models' reported performance on some of the most widely used ones, including:

- NL-code search (Table 11): CodeSearchNet (Husain et al., 2019), CosQA (Huang et al., 2021), and AdvTest (Lu et al., 2021), all measured by Mean Reciprocal Rank (MRR);

- Clone detection (Table 12): BigCloneBench (Svajlenko et al., 2014), measured by F1, and POJ-104 (Mou et al., 2016), measured by Mean Average Precision; the train/test splits of both two datasets are provided by Lu et al. (2021);

- Defect detection (Table 13): Devign (Zhou et al., 2019), measured by accuracy; the train/test split is provided by Lu et al. (2021);

- Code summarization (Table 14): CodeSearchNet (Husain et al., 2019), measured by BLEU; the train/test split is provided by Lu et al. (2021);

- Code translation (Table 15, 16): CodeTrans (Lu et al., 2021), measured by BLEU and CodeBLEU, and Transcoder (Rozière et al., 2020), measured by pass@1;

- Code reasoning (Table 17, 18): CRUXEval (Gu et al., 2024), measured by pass@1, and PythonIO, measured by accuracy.

Table 11: Mean Reciprocal Rank of NL-code search performance on CodeSearchNet, AdvTest, and CosQA.

| | Ruby | JS | Go | Python | Java | PHP | Average | AdvTest | CosQA | source |
|---|---|---|---|---|---|---|---|---|---|---|
| RoBERTa | 58.7 | 51.7 | 85.0 | 58.7 | 59.9 | 56.0 | 61.7 | 18.3 | 60.3 | Lu et al. (2021)↓ |
| CodeBERT | 67.9 | 62.0 | 88.2 | 67.2 | 67.6 | 62.8 | 69.3 | 27.2 | 65.7 | |
| GraphCodeBERT | 70.3 | 64.4 | 89.7 | 69.2 | 69.1 | 64.9 | 71.3 | 35.2 | 68.4 | Guo et al. (2021a) |
| SynCoBERT | 72.2 | 67.7 | 91.3 | 72.4 | 72.3 | 67.8 | 74.0 | 38.1 | 69.6 | Wang et al. (2021d) |
| SPT-Code | 70.1 | 64.1 | 89.5 | 69.9 | 70.0 | 65.1 | 71.5 | | | Niu et al. (2022) |
| CodeRetriever | 77.1 | 71.9 | 92.4 | 75.8 | 76.5 | 70.8 | 77.4 | 46.9 | 75.4 | Li et al. (2022e) |
| UniXcoder | 74.0 | 68.4 | 91.5 | 72.0 | 72.6 | 67.6 | 74.4 | 41.3 | 70.1 | Guo et al. (2022)↓ |
| PLBART | 67.5 | 61.6 | 88.7 | 66.3 | 66.3 | 61.1 | 68.5 | 34.7 | 65.0 | |
| CodeT5$_{base}$ | 71.9 | 65.5 | 88.8 | 69.8 | 68.6 | 64.5 | 71.5 | 39.3 | 67.8 | |
| CoCoSoDa | 81.8 | 76.4 | 92.1 | 75.7 | 76.3 | 70.3 | 78.8 | | | Shi et al. (2023b) |
| Code-MVP | | | | | | | | 40.4 | 72.1 | Wang et al. (2022e) |
| SCodeR | 77.5 | 72.0 | 92.7 | 74.2 | 74.8 | 69.2 | 76.7 | 45.5 | 74.5 | Li et al. (2022f) |
| CodeT5+$_{large}$ | 78.0 | 71.3 | 92.7 | 75.8 | 76.2 | 70.1 | 77.4 | 44.7 | 74.0 | Wang et al. (2023h)↓ |
| CodeGen-multi 350M | 66.0 | 62.2 | 90.0 | 68.6 | 70.1 | 63.9 | 70.1 | 34.8 | 64.8 | |

Table 12: Clone detection performance on BigCloneBench (F1) and POJ-104 (Mean Average Precision). Works in the two blocks use different implementations and report different baseline scores.

|  | BigCloneBench (F1) | POJ-104 (MAP) | source |
|---|---|---|---|
| RoBERTa | 94.9 | 79.96 | Lu et al. (2021)↓ |
| CodeBERT | 96.5 | 84.29 |  |
| DOBF | 96.5 |  | Lachaux et al. (2021) |
| PLBART | 97.2 |  | Ahmad et al. (2021) |
| GraphCodeBERT | 97.1 | 85.16 | Wang et al. (2021d)↓ |
| SynCoBERT | 97.4 | 88.24 |  |
| CodeT5$_{base}$ | 97.2 |  | Wang et al. (2021f) |
| RoBERTa | 91.3 | 76.67 | Guo et al. (2022) |
| CodeBERT | 94.1 | 82.67 | Guo et al. (2021a)↓ |
| GraphCodeBERT | 95.0 | 85.16 |  |
| DISCO | 94.4 | 83.32 | Ding et al. (2022a) |
| PLBART | 93.6 | 86.27 | Guo et al. (2022)↓ |
| CodeT5$_{base}$ | 95.0 | 88.65 |  |
| UniXcoder | 95.2 | 90.52 |  |
| SCodeR | 95.3 | 92.45 | Li et al. (2022f) |

Table 13: Accuracy of defect detection on Devign. pt: pretrained. ft: finetuned.

|  | Accuracy | pt | ft | prompt | source |
|---|---|---|---|---|---|
| Transformer | 61.6 |  | ✓ | - | Ahmad et al. (2021) |
| RoBERTa | 61.0 | ✓ | ✓ | - | Lu et al. (2021)↓ |
| CodeBERT | 62.1 | ✓ | ✓ | - |  |
| PLBART | 63.2 | ✓ | ✓ | - | Ahmad et al. (2021) |
| GraphCodeBERT | 63.2 | ✓ | ✓ | - | Wang et al. (2021d)↓ |
| SynCoBERT | 64.5 | ✓ | ✓ | - |  |
| CodeT5$_{base}$ | 65.8 | ✓ | ✓ | - | Wang et al. (2021f) |
| DISCO | 64.4 | ✓ | ✓ | - | Ding et al. (2022a) |
| VulBERTa | 64.8 | ✓ | ✓ | - | Hanif & Maffeis (2022) |
| Instruct-CodeGen 16B | 47.8 | ✓ |  | 0-shot | Yuan et al. (2023a)↓ |
| CodeAlpaca 7B | 51.9 | ✓ |  | 0-shot |  |
| Vicuna 7B | 54.0 | ✓ |  | 0-shot |  |
| WizardCoder 15B | 54.4 | ✓ |  | 0-shot |  |

Table 14: BLEU scores of code summarization on CodeSearchNet, using the CodeXGLUE split (top) and the original split (bottom). pt: pretrained. ft: finetuned.

| | Ruby | JS | Go | Python | Java | PHP | Average | pt | ft | prompt | source |
|---|---|---|---|---|---|---|---|---|---|---|---|
| Transformer | 11.18 | 11.59 | 16.38 | 15.81 | 16.26 | 22.12 | 15.56 | | ✓ | - | Lu et al. (2021)↓ |
| RoBERTa | 11.17 | 11.90 | 17.72 | 18.14 | 16.47 | 24.02 | 16.57 | ✓ | ✓ | - | |
| CodeBERT | 12.16 | 14.90 | 18.07 | 19.06 | 17.65 | 25.16 | 17.83 | ✓ | ✓ | - | |
| PLBART | 14.11 | 15.56 | 18.91 | 19.30 | 18.45 | 23.58 | 18.32 | ✓ | ✓ | - | Ahmad et al. (2021) |
| CodeT5$_{base}$ | 15.69 | 16.24 | 19.76 | 20.36 | 20.46 | 26.09 | 19.77 | ✓ | ✓ | - | Wang et al. (2021f) |
| UniXcoder | 14.87 | 15.85 | 19.07 | 19.13 | 20.31 | 26.54 | 19.30 | ✓ | ✓ | - | Guo et al. (2022) |
| InCoder | | | | 18.27 | | | | ✓ | | 0-shot | Fried et al. (2023) |
| NatGen | 15.38 | 16.00 | 19.43 | 20.09 | 20.38 | 26.00 | 19.55 | ✓ | ✓ | - | Chakraborty et al. (2022a) |
| CodeT5+$_{large}$ | 15.63 | 17.93 | 19.64 | 20.47 | 20.83 | 26.39 | 20.15 | ✓ | ✓ | - | Wang et al. (2023h) |
| StarCoder | | | | 21.99 | | | | ✓ | | 0-shot | Li et al. (2023i) |
| ChatGPT | | | | 10.28 | | | | ✓ | | 0-shot | Sun et al. (2023b) |
| PaLM 2 | | | | 19.23 | | | | ✓ | | few-shot | Haldar & Hockenmaier (2024)↓ |
| LLaMA 2 70B | | | | 22.41 | | | | ✓ | | few-shot | |
| Mastropaolo et al. | 7.8 | 9.0 | 15.3 | 9.7 | 13.5 | 18.4 | 12.3 | ✓ | ✓ | - | Niu et al. (2022)↓ |
| CugLM | 7.4 | 10.0 | 17.0 | 11.4 | 13.2 | 17.8 | 12.8 | ✓ | ✓ | - | |
| TreeBERT | 7.4 | 10.3 | 17.1 | 11.1 | 13.8 | 18.0 | 12.9 | ✓ | ✓ | - | |
| CodeBERT | 8.3 | 10.0 | 16.0 | 10.7 | 13.6 | 20.1 | 13.1 | ✓ | ✓ | - | |
| GraphCodeBERT | 8.3 | 10.9 | 16.5 | 11.0 | 14.5 | 20.0 | 13.5 | ✓ | ✓ | - | |
| SPT-Code | 8.4 | 12.8 | 18.8 | 12.8 | 16.8 | 20.4 | 15.0 | ✓ | ✓ | - | |

Table 15: BLEU and CodeBLEU scores of code translation on CodeTrans.

| | Java→C# | | C#→Java | | source |
|---|---|---|---|---|---|
| | BLEU | CodeBLEU | BLEU | CodeBLEU | |
| Transformer | 55.8 | 63.7 | 50.5 | 61.6 | Lu et al. (2021)↓ |
| CodeBERT | 79.9 | 85.1 | 72.1 | 79.4 | |
| GraphCodeBERT | 80.6 | | 72.6 | | Guo et al. (2021a) |
| PLBART | 83.0 | 87.9 | 78.3 | 85.3 | Ahmad et al. (2021) |
| SynCoBERT | 80.8 | 84.8 | 76.5 | 82.2 | Wang et al. (2021d) |
| CodeT5$_{base}$ | 84.0 | 87.8 | 79.9 | 84.4 | Wang et al. (2021f) |
| NatGen | | 88.1 | | 85.2 | Chakraborty et al. (2022a) |

Table 16: Pass@1 scores of code translation on TransCoder evaluation set.

| | C++ → Java | C++ → Py | Java → C++ | Java → Py | Py → C++ | Py → Java | source |
|---|---|---|---|---|---|---|---|
| TransCoder | 60.9 | 44.5 | 80.9 | 35.0 | 32.2 | 24.7 | Rozière et al. (2020) |
| DOBF | | | | 40.6 | | 46.6 | Lachaux et al. (2021) |
| TransCoder-ST | 66.7 | 61.1 | 84.1 | 67.8 | 52.2 | 56.7 | Rozière et al. (2022) |
| TransCoder-IR | 62.9 | | 74.5 | | | | Szafraniec et al. (2023) |
| LaMDA | | 30.2 | | | | | Chowdhery et al. (2023)↓ |
| PaLM | | 51.8 | | | | | |
| PaLM Coder | | 55.1 | | | | | |
| StarCoder (self-debug) | | 70.0 (76.6) | | | | | Chen et al. (2023f)↓ |
| Codex (self-debug) | | 80.4 (92.5) | | | | | |
| GPT-3.5 (self-debug) | | 89.1 (92.7) | | | | | |
| GPT-4 (self-debug) | | 77.3 (90.4) | | | | | |

Table 17: Pass@1 scores of code reasoning on CRUXEval input and output prediction. DeepSeek-Coder-V2 is a Mixture-of-Experts model with a total of 236B parameters, where 21B are activated for each token.

| | size | input | output | source |
|---|---|---|---|---|
| Phi-1 | 1.3B | 13.9 | 23.3 | Gu et al. (2024)↓ |
| Phi-1.5 | 1.3B | 24.1 | 27.1 | |
| Mistral | 7B | 36.0 | 31.7 | |
| StarCoder$_{Base}$ | 16B | 31.6 | 33.3 | |
| CodeLLaMA (CoT) | 34B | 46.5 (50.4) | 41.1 (46.0) | |
| DeepSeek-Coder$_{Instruct}$ | 33B | 47.4 | 44.0 | |
| GPT-3.5 (CoT) | | 49.2 (49.1) | 50.0 (63.3) | |
| GPT-4 (CoT) | - | 67.1 (74.8) | 63.4 (81.9) | |
| StarCoder2 | 16B | 48.1 | 47.1 | Lozhkov et al. (2024) |
| Codestral + CoT | 22B | 48.0 | 60.6 | Zhu et al. (2024)↓ |
| LLaMA-3$_{Instruct}$ + CoT | 70B | 61.1 | 64.3 | |
| DeepSeek-Coder-V2$_{Instruct}$ + CoT | 236B (21B) | 70.0 | 75.1 | |
| Gemini-1.5-Pro + CoT | - | 67.0 | 77.5 | |
| Claude-3-Opus + CoT | - | 73.4 | 82.0 | |
| GPT-4o + CoT | - | 77.4 | 88.7 | |

Table 18: Accuracy of code reasoning performance on PythonIO, extracted from Zhang et al. (2024d).

| | size | accuracy |
|---|---|---|
| Phi-2 | 2.7B | 29.8 |
| Phi-3 | 3.8B | 38.2 |
| ChatGLM3$_{Base}$ | 6B | 27.7 |
| Gemma | 7B | 30.5 |
| Mistral | 7B | 31.7 |
| Qwen1.5 | 7B | 32.8 |
| LLaMA-3$_{Instruct}$ | 8B | 39.0 |
| Flan-T5 | 11B | 29.3 |
| LLaMA-2 | 13B | 26.6 |
| Qwen1.5 | 14B | 40.9 |
| LLaMA-2 | 70B | 38.2 |
| LLaMA-3$_{Instruct}$ | 70B | 70.1 |
| Qwen1.5 | 72B | 50.4 |

