# OpenReview forum: "Unifying the Perspectives of NLP and Software Engineering: A Survey on Language Models for Code"
_TMLR — Accepted by TMLR_

### Review · Reviewer_ntKZ · 2024-04-23

**Summary Of Contributions:**

In this paper, the authors provide a fairly deep survey of the use of transformer-based neural networks (mostly large language models (LLMs) in the context of software code. The authors also visually illustrate the trends of advancement using LLMs for code-related tasks. The paper considers over 50 models, 30 evaluation tasks, 170 datasets and 700 related works (the references alone are ~75 pages of the 100 page manuscript).

In addition to the survey, the authors provide some insight on emerging research and engineering directions, including how well LLMs are performing in certain tasks and a categorization (i.e., taxonomy) of natural language and programming language analyses.

**Audience:**

Yes

**Broader Impact Concerns:**

Getting the language right on NL to PL is important, in my opinion, because I think getting it wrong could confuse many readers into thinking LLMs are handling all SE tasks, including those that have nothing to do with code/text. I think not only would that be a disservice to the larger community, but also to the authors and the strong work they’ve put together here.

**Claims And Evidence:**

Yes

**Requested Changes:**

There are two main areas where I think the paper needs improvement before I can give it an accept rating for publication. After reading the paper in its entirety, I have a strong level of confidence the authors will be capable of handling these two items (although they may be non-trivial). The two items are listed below.

1. I think this paper is conflating SE with PL. In almost all cases, the paper is focusing on PL (i.e., analysis or synthesis of code). Yet, the paper is dressing the PL up like it’s SE. However, what is really happening is the authors are documenting the transition of using LLMs from NL to PL. Therefore, the focus is on PL and only the concepts in SE that are *related* to PL (or NL). The paper is not (to my knowledge) focusing on the broader space of SE, but right now there is a lot of confusion in the paper indicating otherwise.

I think this is extremely important to get right. The reason for this is that the field of SE is much, much bigger than just PL. As such, when the authors are saying it’s “NL to SE” many software engineering tasks will need to be covered otherwise the work will miss critical aspects that are pertinent to SE. But I don’t think that’s what the authors are actually *trying* to say. I think what the authors are really intending to say is the following: “this paper is about the use of LLMs for PL, where the PL usage has some potential value in SE.” This is what I think the paper is really about, but the authors appear to be saying “NL to SE” which created a lot of confusion for me. Let me explain.

PL is really about the study and use of programming languages. Like NLP before it, I think this paper is principally about how LLMs, which were initially targeted for NLP, are faring in PLP. SE is really about how to engineer software. Many of those tasks have little or nothing to do with code: requirements, team organization, hiring, scheduling, software dev methods, version control systems, defect tracking, integration testing, software supply chain, dependency analysis, software deployment, etc. As such, after reading the paper, it seems in almost all cases where the authors are talking about SE they are talking principally about how PL can be used for software, not how LLMs cover all aspects of SE (otherwise there are large holes of aspects in SE that are absent).

I think the authors should try to reframe this so it’s clear that the emphasis is on evolution of LLM application for PL in addition to NL. If the authors continue to try to draw the corollary of “NL -> SE” I think we will ultimately be creating more confusion in the community because NLP/NLU is about natural language processing and understanding and software engineering is about a bunch of things that are not related to PL. But NL <-> PL, while there are still major differences, is at least more of an apples to apples comparison, in my opinion (moreover, it appears this is what is actually being discussed in the paper).

2. Some of the organization of the ideas appear to miss perhaps minor, but critical, details that I think may confuse readers if they remain absent or the paper isn’t changed to minimally inform the reader on the broader scope of work.

There are at least three examples of this in the paper: (1) the “code-to-text” categories, (2) “text-to-SQL” being a special case (I’m not convinced that it is) and (3) the organization of the downstream tasks appears less systematic and organized than other aspects. Let’s dissect each one.

“Code-to-text”:

First, I think this gets back to the larger issue of not using the right language and therefore a bit (or a lot) of the message is lost. For example, all code is text, but not all text is code. So if you already have code, then you already have text. So then what are we doing if we translate from “code-to-text”? Code *is* text.

Well, what the paper is *really* trying to say is we are translating PL-to-NL (but because it doesn't say this, unless you know this already, you may become very confused). Moreover, in the other cases we are doing NL-to-PL. Sometimes it’s PL-to-PL and NL-to-NL. But words matter and I think the words “code” and “text” are not the right ones here. Moreover, once we see it’s a NL-to-PL thing it sheds even more light on larger issues: which NLs are going to which PLs and which ones aren’t? But if we use “text-to-code” suddenly it pluralizes the discussion and the hidden details may be lost.

Moreover, by using NL and PL, this opens up the possibility to fix the next major issue which is the lack of mention of specification in the space of software engineering. Specs, unlike generally ambiguous NL are more precise and attempt to describe the “what the software will do” or “what the PL constructs are intended to do.” It is imperative that specs do not describe “how it will be done”, which is why specs (let’s call them specification languages, or SLs) are not written casually in NL nor are they written precisely in PL. In fact, I would argue there is a third middle ground here of NL <> SL <> PL. And once we correct the language mistakes, it becomes more clear that SL fits in quite nicely with the correct nomenclature.

“Text-to-SQL”:

I’m not sure this is a one-off as the paper suggests. I actually think it’s more about SQL being the closest to an intentional programming language (except for perhaps Halide) than any other language. In short, SQL is declarative and declarative languages generally only specify the “what” not the “how” thus it’s far easier to translate NL to declarative/intentional languages than NL to imperative languages (like C, C++). Moreover, the grammar is significantly more restricted in SQL than other languages, like C, C++. Therefore the search is computationally more efficient. I think what is really being described here is: “NL-to-intentional PL” and again, using the right language helps us see the right picture that the PL we’re translating to matters, but because of the type of language and its constraints.

Recommend reading JRK’s research at MIT on Halide for a deeper examination on this to understand this phenomenon does not relate only to SQL. In mine (and others opinion) it’s really about opportunities using intentional languages (“Three Pillars of MP”) as Ryan Marcus et al. discuss in their Bao and Neo SQL research (https://arxiv.org/abs/2004.03814, https://arxiv.org/abs/1904.03711) and why there are so many advances to be made in their inventiveness and adaptation for optimization. It's really about SQL's declarative nature that makes this so well aligned to optimization. The same I believe is true about NL-to-SQL, it's more about "NL-to-intentional PLs".

Downstream tasks:

I liked the list of the downstream tasks for this work, but I couldn’t understand why these were selected nor if they were “exhaustive” in some sense. I think it would help quite a bit to organize the downstream tasks in some kind of framework. Right now, it appears like a long list of things (all of which are important in their own right), but I don’t know how to categorize them in my head.

Sadly, in being transparent, I don’t know how to organize these items (unlike my suggestion of changing “text-to-code” to “NL-to-PL”). My hope is that the authors can understand the concern and work to improve it. I have a strong degree of confidence based on their treatment of the other visual diagrams in the paper that they’ll be able to rectify this elegantly.

**Strengths And Weaknesses:**

Overall, I like this paper. I think it is on its way to being a publishable work. However, I do have some large concerns that may require the authors to rethink some core principles of the paper. Based on the work that the authors have already done, I have confidence that these requests are not too far afield for them to push this paper across the publication-worthy threshold.

Strengths:
- The surveyed work is notable and fairly comprehensive. The authors have clearly spent a reasonable amount of time conducting it and trying to make sense of the many works they analyze.
- I think the trend analysis graphs (Figure 2 and Table 3 (pg 19)) are quite insightful in the important data they provide.
- The “code” tree system the authors use throughout the paper (Figure 1 and then many variants of Figure 3) are extremely helpful to visually digest a large amount of information in a fairly organized way.
- I think the authors provide a reasonable snapshot of the “LLM for code” landscape and its evolution over the last few years.


Weaknesses:
- I am confused about the authors’ use of “software engineering” throughout this paper. To me, it seems that the paper mostly discusses tasks that go from natural language (NL) to programming language (PL). NL <-> PL. Software engineering (SE) has many activities that are entirely outside the scope of PL. For example, Agile Software Development has nothing to do with PL, but has a lot to do with SE practices. The same could be said about software requirements creation; it is a fundamental step of SE but rarely (and usually should not) includes code (PL). One could argue that the process of doing performance profiling has more to do with proper use of perf. tooling, sampling, and heterogenous hardware evaluation than code. The development of UI/UX systems which are fundamental to SE rarely consider code in their design. And so forth … As such, it feels like many aspects of the paper are confused because of the perhaps misappropriation of the term SE where instead the target is PL (principally around “code”).
- I’m not sure I agree with some of the categorization of ideas in the paper (e.g., text-to-SQL being considered a “one-off” – I explain more below, but I believe this is a declarative/intentional programming language thing not a one-off (e.g., Halide has similar SQL-like properties))
- In Section 2, I think there are at least three fundamental aspects that should be resolved. First, I think it’s a misnomer to call these text-to-code and code-to-text categories, as “all code is text, but not all text is code.” So how does one do “code-to-text” translation as code is already text? I think the authors mean to say PL-to-NL; this goes back to the core conflation of PL with SE, in my opinion.  Second, there is an important nuanced space between NL and PL which is the space of “software specifications”, which are neither normal NL and not PL, yet it exists all throughout software. It has the appearance of being NL, but it’s not as it’s more rigorous (constrained) than typical NL, yet it is intentionally not written in PL because then the “spec” would be an “implementation” which it is precisely the opposite its goal as a spec. Third, the downstream items seem pretty disorganized to me. It felt more like a potpourri and an arbitrary selection of topics rather than an organized, rigorous framework for tasks. (For example, the code-to-text and text-to-code is more rigorous because we can see the exhaustiveness emerge, but with the downstream tasks that’s absent). Because of the lack of systematic organization / framework, it’s rather hard to know things like: (1) is this an exhaustive list and (2) is this a proper way to reason about these systems? More on this below.

---

### Review · Reviewer_uFxT · 2024-06-18

**Summary Of Contributions:**

This paper presents a survey of code processing methods using large language models. The paper explores related work in this field by surveying a range of models, datasets and tasks. As opposed to previous surveys, the paper proposes to bridge the gap between tasks coming from NLP and tasks proposed from a software engineering perspective, and surveys tasks and approaches from both perspectives. First, it surveys tasks coming from software engineering (sections 2.1.1 - 2.1.5). Then, it surveys tasks in NLP (section 2.1.6). In sections 4 and 5, the paper surveys general and specialized language models for coding. In section 6 the paper surveys some additional features added to language models for coding tasks. Finally, in section 7 the paper surveys advances of large language models in the software engineering process. Along these sections, the paper presents analyses of existing datasets, metrics and performance of state-of-the-art models for coding.

**Audience:**

Yes

**Broader Impact Concerns:**

No broader impact concerns.

**Claims And Evidence:**

Yes

**Requested Changes:**

Critical changes that would impact recommendation for acceptance:
1. Add experimental results for all or some of the SE tasks presented in 2.1.1 - 2.1.5, or clarify which of the experiments presented in sections 4 and 5 would cover these.

Changes that would strengthen the work:
1. Make the 'unifying perspective' of SE and NLP/modelling perspectives of language models for coding clearer.
2. Improve readability of figures in sections 2.1.1 - 2.1.5.

**Strengths And Weaknesses:**

Strengths:
- The paper presents an interesting survey that aims at closing a gap between efforts to use/evaluate language models from a software engineering perspective and the same efforts from an NLP/modelling perspective. The tasks surveyed in section 2 provide a useful framework to analyse language models from a perspective that could be more useful to software engineering.
- The paper examines a vast amount of research from the two perspectives, which is according to what's expected for a survey, and adds experimental results to at least one of these perspectives which helps strenghten the contribution.

Weaknesses:
- Although I appreciate how systematically the paper presents existing work, and the effort on trying to bridge the gap between SE and NLP perspectives on language models for coding, the organization of the paper seems a bit disconnected from this intention. It seems like most of the content is related to the NLP/modelling perspective, while the SE perspective is only tangentially touched in section 2 by listing methods that perform each of these tasks. I would have expected that, after each perspective is presented (as in section 2), the existing language models would be analysed to the lens of both perspectives (in sections 4 and 5). Or, at least, a more clear "unifying perspective", as claimed in the title of the paper.
- Related to the previous item: considering the thorough analyses regarding the performance of general and specialized language models presented in sections 4 and 5, which I assume cover the NLP/modelling perspective, it feels incomplete that the same type of experimental results are not presented for section 2, where specific types of tasks coming from the SE are mentioned. It would be useful to see actual experimental results for at least some of these SE tasks.
- A minor point: readability of figures in section 2.1.1 - 2.1.5 could be improved by adding titles to the list of references presented in the last three columns of these diagrams.

---

### Review · Reviewer_sQxn · 2024-06-21

**Summary Of Contributions:**

The paper presents a comprehensive literature review about the current research on text-code-related problems (e.g. text-to-code generation, code retrieval). The review covers an impressive number of things: 50+ models, 30+ evaluation tasks, 170+ datasets, and 700+ related works. The paper introduces a task-based taxonomy, covering a wide range of tasks requiring text and code.

**Audience:**

Yes

**Broader Impact Concerns:**

No concern

**Claims And Evidence:**

Yes

**Requested Changes:**

I would like to see
- the progress of other task rather than only text-to-code.
- deeper discussion about the impacts of text-code tasks, especially about which tasks we can use LLMs with high confidence, which tasks can be automated, which tasks are still challenging, etc.

**Strengths And Weaknesses:**

## Strengths
- The paper is an impressive work requiring lots of effort reading articles and summarizing, taxonomizing their contents. I found the paper easy to read and understanding; the structure is clear.

- The taxonomy is intuitive and reasonable.

- Figure 2 is very helpful to capture the progress of code-language models.

## Weaknesses
- This is a good paper yet it seems focus mainly on text-to-code's performance with lots of comparisons (figure 2, tab 1,2,4). However, it's difficult to see the progress in other tasks, and also real applications.

---

### Decision · Action_Editor_7DMJ · 2024-09-02

**Recommendation:** Accept as is

**Comment:**

The reviewers found the survey to be a comprehensive resource for a rapidly evolving space. The authors' desire to connect the differing NLP and SE perspectives to this area is especially laudable. While the original submission had a few weaknesses, these issues were addressed in the revisions.

**Audience:**

The paper should be of interest to researchers working on AI for code, a key application of modern language models.

**Claims And Evidence:**

The paper surveys recent advances in software engineering with LLMs. A key objective of the survey is to connect the Software Engineering (SE) and Natural Language Processing (NLP) perspectives on models of code. To this end, it: (i) summarizes a range of approaches to engineering and evaluating code models, and (ii) lays out ways in which LLMs can benefit a broad set of SE tasks, from program development to software testing and analysis to post-deployment maintenance and updates.